# Threshold-dependent negative autoregulation of *PIF4* gene expression optimizes growth and fitness in Arabidopsis

Sreya Das◉⊙, Vikas Garhwal⊙, Krishanu Mondal, Dipjyoti Das◉*,
Sreeramaiah N. Gangappa◉*

Department of Biological Sciences, Indian Institute of Science Education and Research Kolkata, Mohanpur, West Bengal, India

⊙ These authors contributed equally to this work.
* ngsreeram@iiserkol.ac.in (SNG); dipjyoti.das@iiserkol.ac.in (DD)

## Abstract

PHYTOCHROME INTERACTING FACTOR 4 (PIF4) is a vital transcription factor that controls plant growth by integrating environmental signals like light and temperature. Recent studies have shown many upstream regulators, such as HEMERA (HMR), HEAT SHOCK TRANSCRIPTION FACTORS (HSFs), TEOSINTE BRANCHED 1/CYCLOIDEA/PCF 5 (TCP5), and the B-BOX (BBX) proteins, play roles in regulating *PIF4* transcription. However, the role of PIF4 in controlling its own gene expression is unknown. Here, we demonstrate that the *PIF4* undergoes negative autoregulation. We show that *PIF4* promoter activity is higher in the *pif4* mutant but significantly reduced in *PIF4* overexpression transgenic lines. Moreover, CONSTITUTIVE PHOTOMORPHOGENIC 1 (COP1) enhances PIF4 protein stability and promotes PIF4 autoinhibition. However, Phytochrome B (phyB), a photoreceptor that decreases PIF4 stability, inhibits autoinhibition. We further develop a network-based mathematical model incorporating the PIF4 autoinhibition and other key interactions. Our modeling and data analysis reveal that PIF4 autoregulation depends on a threshold of cellular PIF4 concentration. Our model also successfully predicts the hypocotyl growth and *PIF4* promoter activity in various light and temperature conditions. Moreover, we show that the transgenic lines with enhanced PIF4 function negatively influence biomass and yield, irrespective of photoperiod and temperature. Together, the negative feedback of PIF4 dampens its own function and restrains unregulated growth. Our study thus elucidates the mechanisms of how the phyB-COP1/DET1-PIF4 module controls *PIF4* transcription in tune with the endogenous PIF4 level.

**Data availability statement:** All the code and data to fit and simulate the mathematical model can be found on Github with the link https://github.com/krishanuphy/Negative_regulation_of_PIF4/blob/main/README.md All other relevant data are in the manuscript and its supporting information files.

**Funding:** This work is supported by the Govt. of India grants from the Department of Biotechnology (Ramalingaswami Re-entry Fellowship grant, BT/RLF/Re-entry/ 28/2017), Science and Engineering Research Board (start-up research grant, SRG/2019/000446), Anusandhan National Research Foundation (Core Research Grant, CRG/2023/001553), and an intramural grant from IISER Kolkata to S.N.G. Research from D.D. lab is supported by the Science and Engineering Research Board, Govt. of India (EEQ/2023/000551). The funders had no role in study design, data collection and analysis, decision to publish, or preparation of the manuscript.

**Competing interests:** The authors have declared that no competing interests exist.

## Author summary

PIF4, a bHLH transcription factor, is a key regulator of growth in response to diverse environmental cues. Specifically, PIF4 promotes thermosensory growth by targeting many downstream growth and hormone signaling genes. However, the regulatory mechanisms of *PIF4* gene expression remain elusive. Through experiment and mathematical modelling, we show that PIF4 negatively regulates its own transcription above a threshold PIF4 protein concentration, which is dependent on photoperiod and temperature. Elevated levels of endogenous PIF4 protein are associated with reduced *PIF4* promoter activity and lead to increased growth. Additionally, we show that upstream regulators of PIF4, COP1 and phyB, differentially regulate the PIF4 autoinhibition. COP1 promotes PIF4 autoinhibition, while phyB photoreceptor dampens it. Moreover, our study provides evidence that PIF4 autoinhibition affects the expression of downstream genes, regulating biomass and grain yield in response to varying light and temperatures. Together, our study highlights that PIF4-mediated negative feedback is a key regulatory mechanism in controlling growth.

## Introduction

As sessile organisms, plants constantly monitor diurnal and seasonal environmental changes, such as light and temperature, to control their metabolism, growth, and reproduction [1–4]. Temperature and light have antagonistic relationships [5]. Warm temperatures promote growth but compromise plant immunity and seed yield [6–8]. Therefore, balancing these two signaling pathways is critical for optimal growth and fitness [3,5,9].

PHYTOCHROME INTERACTING FACTOR 4 (PIF4) is a central regulator of plant development and integrates light and temperature signals [10–14]. Both light and the circadian clock coordinate to control *PIF4* expression by an external coincidence mechanism [15,16]. *PIF4* mRNA and protein levels are elevated in response to warmth [17,18]. *PIF4* overexpression transgenic lines show a constitutive thermomorphogenic response even under cooler temperatures [18–20]. Therefore, tight control of PIF4 levels within a range is critical for optimal growth and reproduction. Several upstream regulators, such as phytochrome B (phyB), EARLY FLOWERING 3 (ELF3), ELONGATED HYPOCOTYL 5 (HY5), and CONSTITUTIVE PHOTOMORPHOGENIC 1 (COP1)/ DE-ETIOLATED 1 (DET1)/ SUPPRESSOR OF PHYTOCHROME A-105 (SPA), have been shown to control PIF4 activity [12,21]. The phyB degrades PIF4 in a red-light-dependent manner [19,22,23]. ELF3, a component of the evening complex, inhibits *PIF4* transcription and sequesters it by forming heterodimers [24–26]. Similarly, HY5 inhibits the transcriptional capability of PIF4, probably via competitive binding to target promoters [27–29]. On the other hand, COP1/DET1/SPA promotes PIF4 protein stability [29–31]. Moreover, several positive regulators of PIF4 either activate its gene expression or promote protein accumulation. For instance, HEMERA

has been shown to stabilize PIF4 and regulate its target genes, and thermomorphogenesis in Arabidopsis [32]. HSFA1s bind to PIF4 and block its interaction with active phyB Pfr. PIF4 accumulation activates thermoresponsive genes, including *YUCCA8* (*YUC8*), resulting in daytime thermomorphogenesis [33]. Warm temperatures activate HSFA2 expression during the day through HSFA1 regulators, leading to heat shock reactions and thermotolerance [33,34]. TCP5 directly interacts with PIF4 to enhance the stability and activity of the PIF4 protein, but it also directly binds the PIF4 promoter and increases the transcriptional level of PIF4 [35]. TCP5 and PIF4 coactivate numerous temperature-responsive genes to promote hypocotyl and petiole development [35]. Together with clock components CIRCADIAN CLOCK ASSOCIATED 1 (CCA1) and LATE ELONGATED HYPOCOTYL (LHY), the SHORT HYPOCOTYL UNDER BLUE1 (SHB1) directly binds to the *PIF4* promoter to activate its transcription [36]. Moreover, COP1 SUPPRESSOR 4 (CSU4) together with CCA1 regulates *PIF4* expression, wherein CSU4 acts as a transcriptional repressor of CCA1 to inhibit *PIF4* gene expression [37]. The B-BOX (BBX) proteins BBX18 and BBX23 also regulate a subset of PIF4-dependent thermoresponsive genes by inhibiting ELF3 function, emphasizing the significance of BBX18 and BBX23 in thermomorphogenesis [38].

Several transcription factors have evolved autoregulatory mechanisms through feedback or feedforward loops to control their function [39–41]. For instance, HY5, an essential activator of seedling photomorphogenesis, autoactivates to induce its transcription to amplify the signalling output [42,43]. Contrastingly, the basic-helix-loop-helix (bHLH) transcription factor MYC2, a key regulator of growth and insect resistance pathway, negatively autoregulates its gene expression [44]. Also, the transcription factor p53 autoregulates its gene expression both positively and negatively in various developmental contexts [45–47]. PIF4 has been shown to indirectly autoregulate its function by activating the gene expression of its inhibitors [19,48–50]. An unresolved question is whether PIF4 can regulate its own gene expression and, if it does, what type of feedback mechanism might be involved. By combining detailed experimental analysis with mathematical modeling, we show that PIF4 negatively autoregulates its promoter activity when the cellular PIF4 concentration exceeds a threshold. The photoreceptor phyB promotes the *PIF4* promoter activity by reducing PIF4-dependent negative feedback. However, COP1/DET1 promotes PIF4 autoinhibition by increasing PIF4 protein accumulation. Based on these observations, we propose a minimal network model that qualitatively predicts the *PIF4* activity and hypocotyl growth in various environmental conditions and genetic backgrounds. Moreover, we found that negative autoregulation of PIF4 is critical to dampening the expression of its target genes related to growth and hormone signalling, regulating biomass and yield.

## Results

### PIF4 negatively autoregulates its own expression in a photoperiod-dependent manner

To investigate the role of PIF4 in regulating its own expression, we generated two transgenic lines (line #1 and line #2) containing the *PIF4* promoter fused to beta-glucuronidase (GUS) reporter (*pPIF4:GUS*) (S1A Fig) in an *Arabidopsis* wild-type (WT) background. The hypocotyl lengths of these *pPIF4:GUS* transgenic lines seedlings were similar to WT (S1B Fig). We also monitored GUS staining and activity of these lines under short-day (SD) and long-day (LD) photoperiods (S1C-1F Fig). The staining patterns of both lines are largely similar in SD (S1C and S1D Fig), though line #1 showed slightly more activity than line #2 in LD (S1E and S1F Fig). The *pPIF4:GUS* promoter activity peaked both during the day and at the end of the night in SD (S1C and S1D Fig), while in LD, it peaked during the daytime (S1E and S1F Fig), similar to the endogenous *PIF4* transcript accumulation, as reported earlier [51–54]. Also, the GUS activity was significantly higher in the LD than in the SD across different time points over a day (S1G Fig). We randomly chose the transgenic line #1 and introgressed it into the *pif4–101* null mutant (S1H and S1I Fig) and two overexpression lines (*PIF4-OE1* and *PIF4-OE2*) [20], and generated homozygous *pPIF4:GUS* lines under respective genetic backgrounds. The two PIF4-overexpression lines, especially driven by its native promoter, showed that PIF4 protein accumulation is higher than WT, as revealed by immunoblotting data (Fig 1A). *PIF4-OE2* has relatively more PIF4 protein than *PIF4-OE1* (Fig 1A). Consistently, *PIF4-OE2* has longer hypocotyls than *PIF4-OE1* under SD and LD, respectively, albeit both were longer than WT (S1J and S1K Fig). Moreover, qPCR data suggest that the *PIF4-OE1* line has higher *PIF4* transcripts than *PIF4-OE2*,

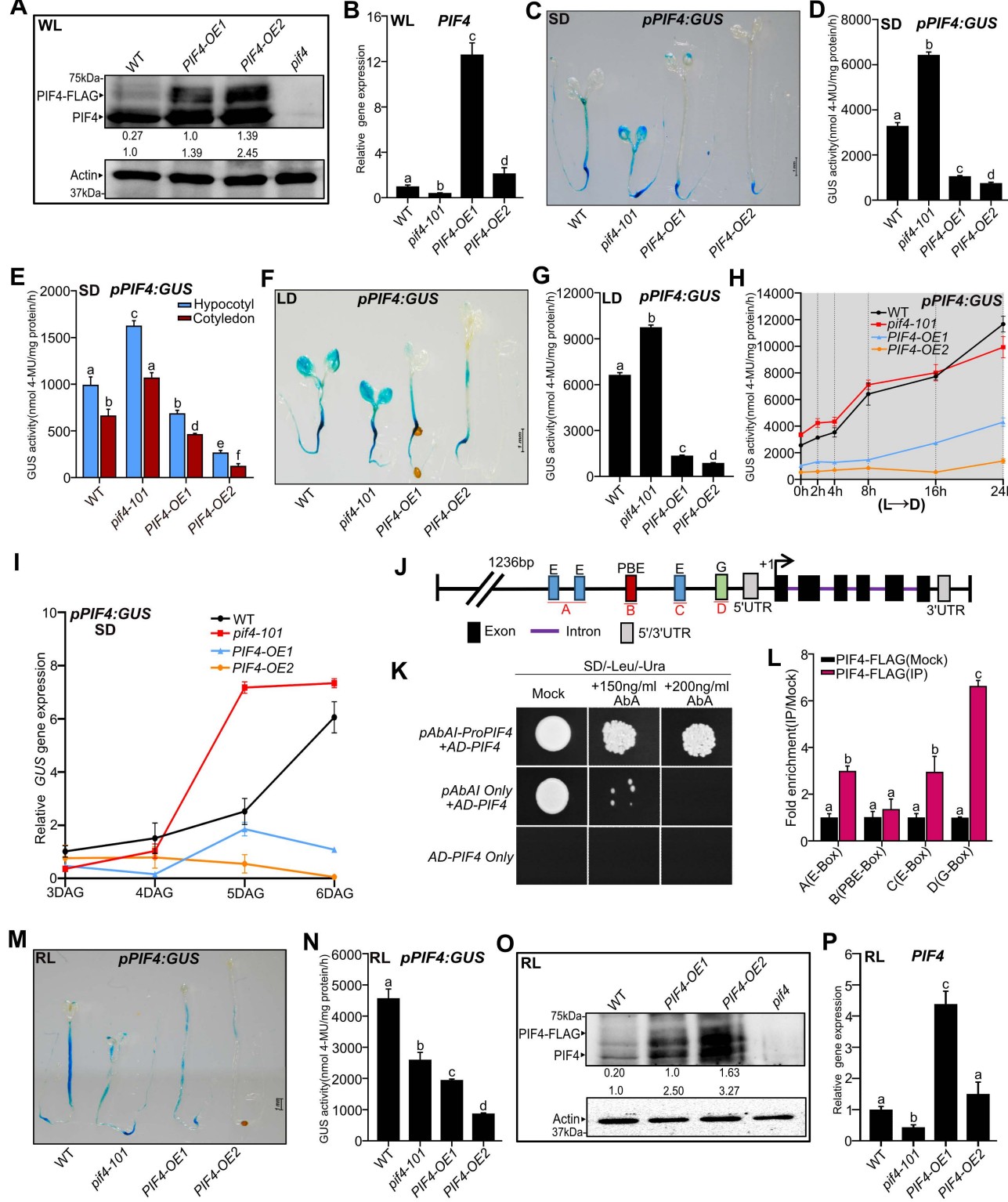

**Fig 1. PIF4 autoinhibits its expression depending on photoperiods. (A)** Immunoblot analysis of WT, *PIF4-OE1* and *PIF4-OE2* using native anti-PIF4 antibody from SD-grown six-day-old seedlings at ZT23 in WL at 22°C. The endogenous PIF4 and PIF4-FLAG protein bands are indicated by arrowheads. Both bands were considered for total PIF4 protein quantification. The total PIF4 protein level below the blots (bottom) was normalized by WT.

Values for PIF4-FLAG (top) indicate the fold change compared to *PIF4-OE1*. Actin was used as a loading control. The *pif4-101* (*pif4*) mutant was used as a negative control. **(B)** Relative *PIF4* transcript accumulation in six-day-old seedlings of respective genotypes grown under SD in WL and tissue harvested at ZT23. **(C and D)** Representative images of GUS staining (C) and quantitative GUS activity measurement (D) from six-day-old whole seedlings of WT, *pif4-101* mutant, *PIF4-OE1 and PIF4-OE2* grown under SD in WL, and seedlings were harvested at ZT23. **(E)** For tissue-specific GUS activity, hypocotyls and cotyledons were excised by scalpel from six-day-old whole seedlings of indicated genotypes grown under SD. Tissue was harvested at ZT23, and GUS activity was quantified. **(F and G)** Representative images of GUS-stained seedlings (F) and quantitative GUS activity (G) from six-day-old seedlings of indicated genotypes grown under LD in WL. **(H)** Genotypes were grown under constant light for five days and shifted into the dark for various durations as indicated, tissue was harvested at the specified intervals, and GUS activity was quantified. The grey zone represents the dark period. **(I)** The seeds were germinated for two days, and *GUS* transcript abundance was measured from three to six-day-old seedlings of WT, *pif4-101, PIF4-OE1* and *PIF4-OE2* grown under SD. Tissue was collected at the end of the night. **(J)** The schematic diagram of the *PIF4* promoter contains the E-box, PBE-box, and G-BOX. **(K and L)** Yeast One-Hybrid assay (K) and ChIP data (L) show that PIF4 binds to its own promoter. The ChIP assay was performed from six-day-old *PIF4-FLAG* seedlings grown under WL-SD (tissue was harvested at ZT23), which shows that PIF4 binds specifically to G-box and E-box **(L)**. **(M and N)** Representative images of six-day-old GUS-stained seedlings (M) and GUS activity (N) in indicated seedlings grown in RL under SD. **(O)** Immunoblot showing endogenous PIF4 and PIF4-FLAG in RL-grown six-day-old seedlings under SD. Tissue was harvested at ZT23. The quantification was done as described in 1A. Actin was used as a loading control. *pif4* was used as a negative control. **(P)** The *PIF4* gene expression at ZT23 from six-day-old seedlings of the indicated genotypes grown in RL under SD. All bar graphs represent data (mean ± SD) from three biological replicates. Relative gene expression was calculated normalizing to the *EF1α*. Different letters indicate a significant difference (one-way and two-way ANOVA with Tukey's HSD test, $P < 0.05$, $n \geq 40$ seedlings for GUS activity data). See also S1-S3 Figs.

though *PIF4-OE1* has less PIF4 protein level than *PIF4-OE2* (Fig 1A and 1B). The GUS staining in the white light (WL)-grown WT seedlings under SD was mainly visible in the hypocotyls and the hypocotyl-root junction (Fig 1C). The *pif4–101* mutant showed a similar staining pattern, but the intensity was more prominent than WT in the cotyledons and the hypocotyl-root junction (Fig 1C). Consistent with this, quantitative GUS activity measurement of entire seedlings was two-fold higher in *pif4–101* than WT (Fig 1D), suggesting PIF4 may negatively regulate its own transcription. On the other hand, in *PIF4-OE1* and *PIF4-OE2*, the GUS staining was absent in the hypocotyls, though *PIF4-OE1* showed mild staining in the cotyledon tips (Fig 1C). Consistently, the quantitative GUS activity from seedlings was several-fold lower than WT in both overexpression lines (Fig 1D). Notably, the GUS activity was significantly reduced in *PIF4-OE2* than in *PIF4-OE1* (Fig 1C and 1D), suggesting that higher endogenous PIF4 concentration correlates with lower *PIF4* promoter activity. We further separately quantified the GUS activity from cotyledons, hypocotyls, and roots. Consistent with the *pPIF4:GUS* promoter activity in whole seedlings, the GUS activity in SD is higher in the *pif4–101* mutant but lower in overexpression lines than in WT in all tissues (Figs 1E and S1L). Moreover, the GUS activity in *PIF4-OE2* is significantly reduced compared to *PIF4-OE1* in all the tissues (Figs 1E and S1L). The *PIF4* promoter activity under LD followed a similar trend to SD in whole seedlings (Fig 1F and 1G) and in different tissues (S1M and S1N Fig). However, in the *PIF4-OE1*, the promoter activity in the hypocotyl was comparable to WT (S1M Fig), and in the *pif4–101* mutant, the activity in roots was similar to WT (S1N Fig). Overall, our data suggest that increased PIF4 function enhances autoinhibition of its promoter activity irrespective of the tissue type. This effect is likely caused by negative feedback regulation of the promoter due to elevated levels of PIF4 protein, which helps sustain homeostasis.

Comparisons between LD and SD further suggest that PIF4 autoinhibition is photoperiod-dependent (S1O and S1P Fig). As already known in the literature [51–54], the PIF4 activity generally increases in the SD than in the LD. Thus, according to our hypothesis of PIF4 autoinhibition, we would expect higher inhibition strength in SD than in LD. Consistent with this, we found lower GUS activity when we compared SD (ZT23) and LD (ZT4) in the WT and *pif4* mutant (S1O Fig). Moreover, at ZT4, we found increased activity in LD than in SD in both WT and *pif4* mutants (S1P Fig). However, the overexpression lines did not show marked differences in activity between SD and LD (S1O and S1P Fig), likely because the PIF4 protein level is already higher in the overexpression lines than in the WT.

Additionally, we shifted five-day-old constant light-grown seedlings to the dark for different exposure times. Increased duration of dark exposure induced GUS activity in WT and *pif4–101* (Figs 1H and S1Q). In contrast, although induced by dark exposure, the promoter activity in the *PIF4-OE1* line was markedly lower than in WT (Figs 1H and SQ1). Moreover, in

the *PIF4-OE2* line, the promoter activity stayed very low and showed no induction by dark treatment (Figs 1H and S1Q). Thus, our data confirms that cellular PIF4 concentration can negatively affect its promoter activity.

We also confirmed the effect of PIF4 on its autoinhibition using *pPIF4:LUC* promoter-reporter [55]. We introgressed the *pPIF4:LUC* into *pif4–101* and *PIF4-OE2* and identified double homozygous lines. Quantification of luciferase (LUC) activity as measured by luminescence from six-day-old seedlings grown in WL under SD photoperiod reveals that the *pPIF4:LUC* promoter activity was significantly upregulated in *pif4–101* but downregulated in *PIF4-OE2* (S1R Fig), further confirming PIF4 autoinhibition.

To further understand *PIF4* autoinhibition linked to protein concentration, we introduced the *pPIF4:GUS* transgene into the *35S:PIF4-HA* transgenic line [19,54], which has exaggerated growth phenotypes [19,54,56]. We compared the hypocotyl phenotype of *35S:PIF4-HA* (referred to as *PIF4-OE3*) with *PIF4-OE1* and *PIF4-OE2* and found that *PIF4-OE3* has longer hypocotyls than *PIF4-OE1* and *PIF4-OE2* (S2A Fig). Moreover, RT-qPCR data using primers specific to *5' UTR* and part of the gene revealed that *PIF4-OE3* has lower endogenous *PIF4* transcript than *PIF4-OE1*, *PIF4-OE2* and WT (S2B Fig). Further, GUS staining in *PIF4-OE3* was hardly visible in any parts of the seedlings, similar to *PIF4-OE2* in both SD and LD (S2C and S2E Fig). Interestingly, the *pPIF4:GUS* promoter activity in *PIF4-OE3* was significantly reduced compared to *PIF4-OE1* and *PIF4-OE2,* both in SD and LD for whole seedlings (S2D and S2F Fig) as well as in different tissues (S2G-S2J Fig).

Similar to seedlings, the *pPIF4:GUS* activity was prominent in the juvenile plants (three-week-old) of both the lines (#1 and #2) grown in WL and showed similar expression patterns and GUS activity (S3A and S3B Fig). Further, analysis of the GUS activity of leaves and stems of six-week-old adult plants showed that the *pif4–101* mutant still has significantly higher GUS activity than overexpression lines, though it has lower activity than WT (S3C-S3H Fig). This suggests that autoinhibition of the *PIF4* promoter persists in the adult stage, albeit to a lesser extent.

Further, to capture the dynamics of negative feedback activation, we quantified the GUS transcripts in the *pPIF4:GUS* line over six days (Fig 1I). We found that the GUS transcript level in the WT steadily increased from day 3 to day 6 (Fig 1I). However, the GUS transcript level in *PIF4-OE1* was significantly lower than the WT at the beginning and slightly increased on the fifth and sixth day, but still stayed markedly lower than WT (Fig 1I). This data is consistent with the GUS activity measured on day 6 (Fig 1D) and suggests that autoinhibition of *PIF4* is active in *PIF4-OE1*. Moreover, the *GUS* transcript level is always lower in *PIF4-OE2* than in WT throughout the duration (Fig 1I), and it is further reduced than *PIF4-OE1* on day 5 and day 6 (Fig 1I). These results suggest that an increased PIF4 accumulation over time results in stronger autoinhibition on the fifth and sixth days.

As our results indicate *PIF4* self-inhibition, we examined whether the PIF4 protein can directly associate with its own promoter. The *PIF4* promoter contains cis-acting elements, such as E-boxes, PBE-box, and G-box, which could be targeted by PIF4 [10] (Fig 1J). When we transformed *pPIF4:AUR1-C* promoter-reporter constructs along with PIF4 as an effector, PIF4 could activate the *AUR1-C* gene expression and the growth of yeast in the presence of Aureobasidin A (AbA), suggesting that PIF4 could bind to its promoter (Fig 1K). Further, Chromatin Immunoprecipitation (ChIP) assay using six-day-old *pPIF4:PIF4-FLAG* transgenic seedlings and subsequent qPCR analysis revealed that PIF4 has more affinity to the G-box than the E-box but not to the PBE-box (Fig 1L).

## Effect of red light on PIF4 autoinhibition

As red light is the vital regulator of the PIF4 function [57], we also compared the GUS staining and *pPIF4:GUS* promoter activity in seedlings grown under red light (RL) and white light (WL). We found higher *PIF4* promoter activity under RL than WL, suggesting differential regulation of *PIF4* activity (S3I and S3J Fig). To further explore how red light influences PIF4-mediated feedback regulation, we measured PIF4 promoter activities in *pif4–101* mutant and *PIF4-OE* lines. We found that GUS staining and activity were reduced in the *pif4–101* mutant than in WT (Fig 1M and 1N), unlike in WL. However, the promoter activity in overexpression lines was significantly reduced compared to WT, similar to WL (Fig 1M

and 1N). Moreover, both *PIF4-OE1* and *PIF4-OE2* have higher *PIF4* proteins than the WT, while the *PIF4* protein level is increased in *PIF4-OE2* than in *PIF4-OE1* (Fig 1O). However, the *PIF4* transcript in *PIF4-OE1* is higher than the WT, though the transcript in *PIF4-OE2* is comparable to the WT (Fig 1P). Thus, both in WL and RL, the PIF4 protein and transcript levels do not have a one-to-one correspondence, likely due to the underlying negative feedback (Fig 1A, 1B, 1O and 1P). This might suggest that the autoinhibition of *PIF4* promoter activity depends on a threshold of PIF4 concentration. For instance, the higher protein level in *PIF4-OE2* than in *PIF4-OE1* (Fig 1A and 1O) leads to stronger inhibition of promoter activity in *PIF4-OE2* than *PIF4-OE1* (Fig 1D and 1N), hence, lower *PIF4* transcripts in *PIF4-OE2* than *PIF4-OE1* (Fig 1B and 1P). To check this hypothesis, we next developed a mathematical model of the threshold-dependent auto-regulation of *PIF4* activity.

**A mathematical model of PIF4 autoinhibition suggests that *PIF4* promoter activity is linked to the endogenous PIF4 threshold**

Based on a previous study [58], we developed a model of PIF4 autoregulation to understand its role in hypocotyl growth (Fig 2A). We assumed that PIF4 autoinhibits its expression when the cellular PIF4 concentration is above a threshold. Our model is based on a minimal genetic network of key regulators, ELF3, phyB, and COP1, which affect PIF4 activity. We incorporated known regulations on PIF4, such as (i) ELF3-dependent repression of *PIF4* transcription, (ii) phyB-dependent inhibition of PIF4 activity, and (iii) COP1-dependent stabilization of PIF4. As reported previously [59,60], we considered the photoactivated form of phyB, which becomes inactive in the dark. Our model also includes light-dependent synthesis of phyB and COP1. We modelled the ELF3 synthesis by an oscillatory function since ELF3 dynamics link the circadian clock with the diurnal cycle [24]. We also incorporated the GUS activity, which is driven by a transgenic promoter-reporter apart from the endogenous *PIF4* promoter (Fig 2A). This transgenic promoter is also negatively regulated by PIF4, dependent on a threshold PIF4 concentration. Finally, we incorporated a variable presenting the hypocotyl growth (measured in mm) as the outcome of PIF4-dependent activation of growth genes (see the detailed mathematical description in Materials and Methods).

We used our model to predict the measured hypocotyl length and GUS activity in different genetic backgrounds. The model was first fitted to wildtype data from seedlings grown under different photoperiods at 22°C to estimate WT parameters [Fig 2B(i), parameters are given in S1 and S2 Tables]. We then quantitatively predicted the hypocotyl lengths in *pif4–101* mutant and overexpression lines by varying distinct synthesis rates of PIF4 in different genotypes (Fig 2B(ii)-(iv)). Here, we assumed that the threshold concentration of PIF4, above which the autoinhibition is active, may vary with temperature and other genetic perturbations (S2 Table). Our theoretical predictions matched well with the hypocotyl data when we assumed the negative feedback on PIF4 (Fig 2B). Notably, without feedback, the predicted hypocotyl lengths are much higher in all genotypes (dashed lines in Fig 2B(i)-(iv)). This indicates that the negative feedback on PIF4 is necessary to limit the unbounded growth within a normal range. Also, our model displayed the oscillatory behaviours of PIF4 concentration and hypocotyl growth (Fig 2C) with the diurnal cycle, as reported previously [58].

Moreover, we simulated the experimental condition where seedlings were grown at a constant WL for five days and then transferred into the dark for one day. Our model prediction is qualitatively similar to the experimental observation that overexpression lines show much lesser GUS activity than the wildtype and *pif4–101* during the long dark treatment (Figs 1H and 2D). However, an exact quantitative comparison of the GUS activity with experimental data cannot be performed since our model's GUS activity is measured in arbitrary units.

Next, we checked the hypothesis of whether a threshold-dependent PIF4 autoinhibition could explain the differential GUS activity observed under WL and RL (Fig 1D and 1N). Note that the relative level of *PIF4* mRNA in the overexpression lines under RL is lower than the WL (Fig 1B and 1P). Thus, according to our hypothesis of PIF4 autoinhibition, we may expect stronger negative feedback in RL than in WL. Therefore, we simulated the WL and RL conditions by increasing the strength of autoinhibition in RL compared to WL. This assumption was sufficient to theoretically reproduce the observed

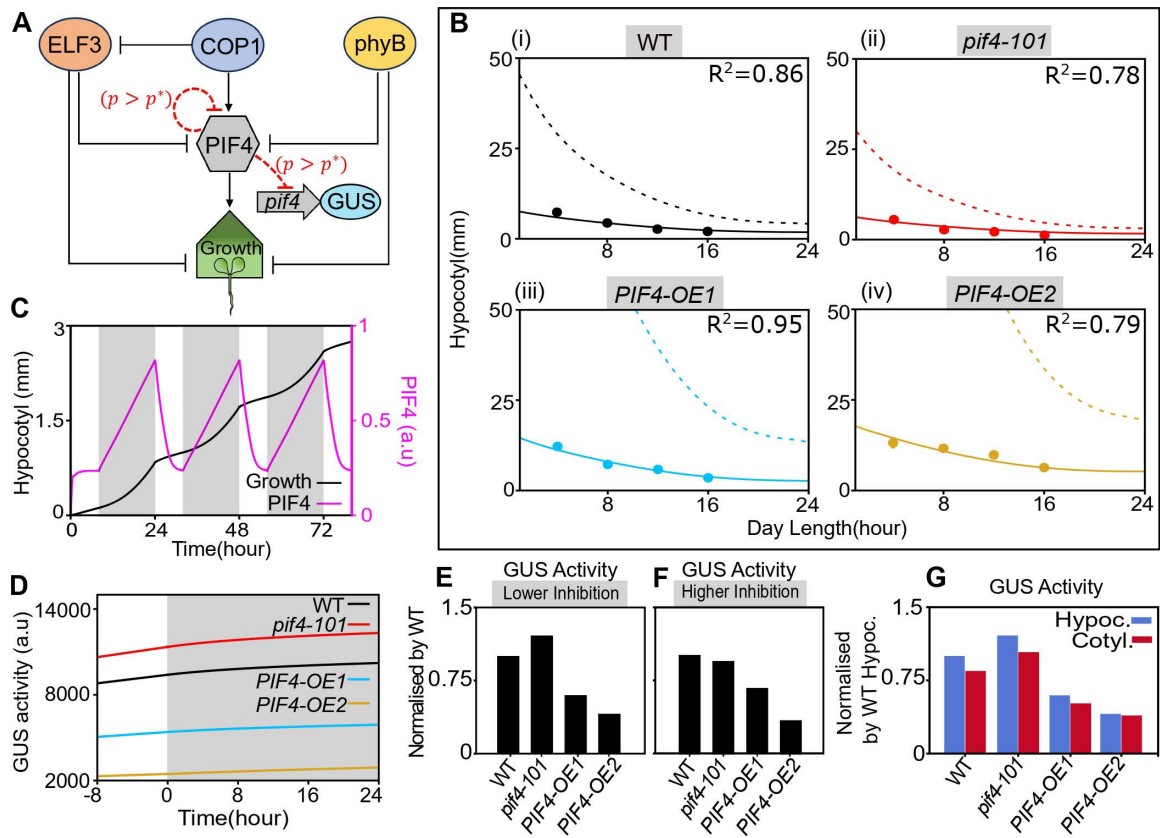

**Fig 2. A mathematical model of PIF4 autoregulation captures the experimental observations of hypocotyl growth and promoter activity. (A)** A schematic network of our model showing the regulations on PIF4. The red dashed lines denote a PIF4 autoinhibition above a threshold PIF4 concentration (see Materials and Methods for details). **(B)** The hypocotyl length versus day length in four genotypes in WL at 22°C. Solid lines represent predictions from our model, assuming PIF4 autoinhibition. Dashed lines denote the predictions without autoinhibition. Filled circles are experimental data for the day lengths 4,8,12, and 16 hours. (n = 20; Error bars represent Standard Deviation). The goodness of fit is shown as $R^2$ values. **(C)** The hypocotyl growth and PIF4 concentration over time, predicted in our model for WT. The bright and shaded regions indicate light and dark periods within a diurnal cycle at 22°C. **(D)** Simulated temporal dynamics of GUS activity for four genotypes during the light-dark transition (also see Fig 1H). The grey zone represents the dark period. **(E, F)** Predicted GUS activity, normalized by the WT value, under SD in WL, assuming either low **(E)** ($P_{self} = 25$) or high **(F)** ($P_{self} = 75$) autoinhibition strength of PIF4 activity. **(G)** Predicted GUS activities in cotyledon (red) and hypocotyl (blue), normalized by the WT value in hypocotyl (compare with Fig 1E). We assumed the autoinhibition strength in cotyledon ($P_{self} = 15$) is lower than in hypocotyl ($P_{self} = 25$). Other model parameters are summarized in S1 and S2 Tables.

differential GUS activity in WL and RL (Fig 2E and 2F). Moreover, our model can also explain the experimentally observed tissue-specific GUS activities in cotyledon and hypocotyl by assuming that the autoinhibition strength in cotyledon is lower than in hypocotyl (compare Figs 1E and 2G). This assumption is based on our data, which shows lower GUS staining in cotyledons than in hypocotyls (Fig 1C). Overall, our model analysis suggests that PIF4 autoinhibits its expression in a threshold-dependent manner.

### The COP1/DET1 module is essential for the *PIF4* gene expression and promotes its autoinhibition

COP1/DET1 module, a master regulator of photomorphogenesis [61–65], promotes PIF4-mediated thermosensory hypocotyl growth by stabilizing the PIF4 protein [28,29,66]. To understand the role of COP1 and DET1 in regulating *PIF4* promoter activity, we introgressed *pPIF4:GUS* transgene into *cop1–4*, *cop1–6*, and *det1–1* mutants and *COP1* overexpression (*35S:COP1*) [67] backgrounds. In *cop1–4* and *cop1–6* seedlings grown in SD photoperiod, the GUS staining

was not visible in any tissues, compared to WT (Fig 3A). In the *det1–1* mutant, the intensity was visibly reduced, though the staining pattern was largely similar to the WT (Fig 3A). Consistently, the GUS activity decreased several-fold in the *cop1–4, cop1–6* and *det1–1* mutants (Fig 3B), implying that COP1 and DET1 positively regulate *PIF4* promoter activity. Therefore, it is expected that increased COP1 activity would increase *pPIF4:GUS* activity. However, surprisingly, the promoter activity was significantly reduced in the *35S:COP1* than in the WT (Fig 3A and 3B). Moreover, qRT-PCR data also suggested that the *PIF4* transcript level was significantly lower than WT in *cop1–4, cop1–6, det1–1* and *35S:COP1* backgrounds (Fig 3C). This unexpected result of reduced GUS activity and *PIF4* transcript in *35S:COP1* can be explained by PIF4 concentration-dependent autoinhibition. By assuming a threshold-dependent PIF4 autoinhibition, our model prediction qualitatively matched the experimental GUS activity (Fig 3D). In line with this, PIF4 protein accumulation in *35S:COP1* was higher than the WT and other mutants (Fig 3E). Like SD, the effect of COP1/DET1 on *pPIF4:GUS* activity was also broadly similar under LD-grown seedlings, as observed in experiments and our model (Fig 3F-3H). Furthermore, the phenotypic output measured by hypocotyl lengths in all genotypes was consistent with our hypothesized model of PIF4 autoinhibition since our prediction and experimental data matched well for both SD and LD (Figs 3I, S4A and S4B). Moreover, constant light-grown WT seedlings, exposed to dark for various durations, showed strong induction of p*PIF4:GUS* promoter activity (as shown above), but the activity was compromised in the *cop1–4, cop1–6* and *det1–1* mutants (Fig 3J and 3K). However, in the *35S:COP1* transgenic line, the promoter activity, although induced more than the mutants, is significantly less than the WT (Fig 3J and 3K).

To explore how COP1 influences *PIF4* promoter activity, we further examined the COP1 protein and transcript levels in *cop1* mutants (*cop1–4, cop1–6*) and *35S:COP1* crossed with *pPIF4:GUS* (S4C and S4D Fig). We found that the *COP1* mRNA level in *cop1–6* was similar to WT as reported before [68], whereas the *COP1* expression was slightly reduced in the *cop1–4* mutant, and significantly increased in *35S:COP1* compared to WT (S4C Fig). Further, the immunoblot analysis using native COP1 antibody showed more protein accumulation in *35S:COP1* compared to WT (S4D Fig). Moreover, in both *cop1–4* and *cop1–6,* COP1 protein levels were markedly reduced (S4D Fig). This suggests that reduced COP1 activity in the mutants results in reduced *PIF4* promoter activity, as expected (Fig 3A and 3B). However, elevated COP1 activity in the *35S:COP1* also reduces promoter activity, likely due to higher PIF4 protein accumulation and corresponding feedback inhibition (Fig 3B and 3E).

Since COP1 positively regulates *PIF4* promoter activity (Fig 3B and 3C), we expect that there should be a threshold PIF4 concentration above which PIF4-mediated autoinhibition dominates over COP1-dependent positive regulation. To test this, we examined the GUS staining and *PIF4* promoter activity in three to six-day-old seedlings in *35S:COP1* (S4E and S4F Fig). Interestingly, the *PIF4* promoter activity in *35S:COP1* was comparable to that of the wild type on day 3, and then the promoter activity in *35S:COP1* became systematically reduced compared to wild-type (S4F Fig). Thus, our data suggest that though the *PIF4* promoter activity in *35S:COP1* was similar to wild-type at early stages, subsequently, the PIF4 level quickly goes above the threshold due to elevated COP1 level in *35S:COP1*, likely leading to stronger PIF4 autoinhibition at later stages.

Like white light, the effect of COP1/DET1 on GUS staining was roughly similar under red light SD-grown seedlings (Fig 3L), and the measured GUS activity data also qualitatively matched with the model (Fig 3M and 3N). Furthermore, the GUS staining and activity of juvenile (S4G and S4H Fig) and adult plants (S4I-S4L Fig) revealed that COP1/DET1 positively regulates *pPIF4:GUS* activity irrespective of tissue type and developmental stages.

The circadian clock components CCA1 and LHY, together with SHB1 activate *PIF4* expression [36]. We explored whether COP1-mediated regulation of *PIF4* promoter activity is dependent on CCA1/LHY and SHB1 activity. Thus, we performed gene expression analysis for *CCA1, LHY* and *SHB1* and western blot for CCA1 in *cop1–4, cop1–6* mutants and *35S:COP1*. The transcript levels of *CCA1* and *LHY* were significantly elevated in the *cop1* mutant compared to WT (S4M and S4N Fig). Interestingly, *CCA1* and *LHY* transcript levels were also increased in the *35S:COP1* compared to WT (S4M and S4N Fig). Similarly, the *SHB1* transcript level in the *cop1–6* mutant and *35S:COP1* was elevated, but in the *cop1–4*

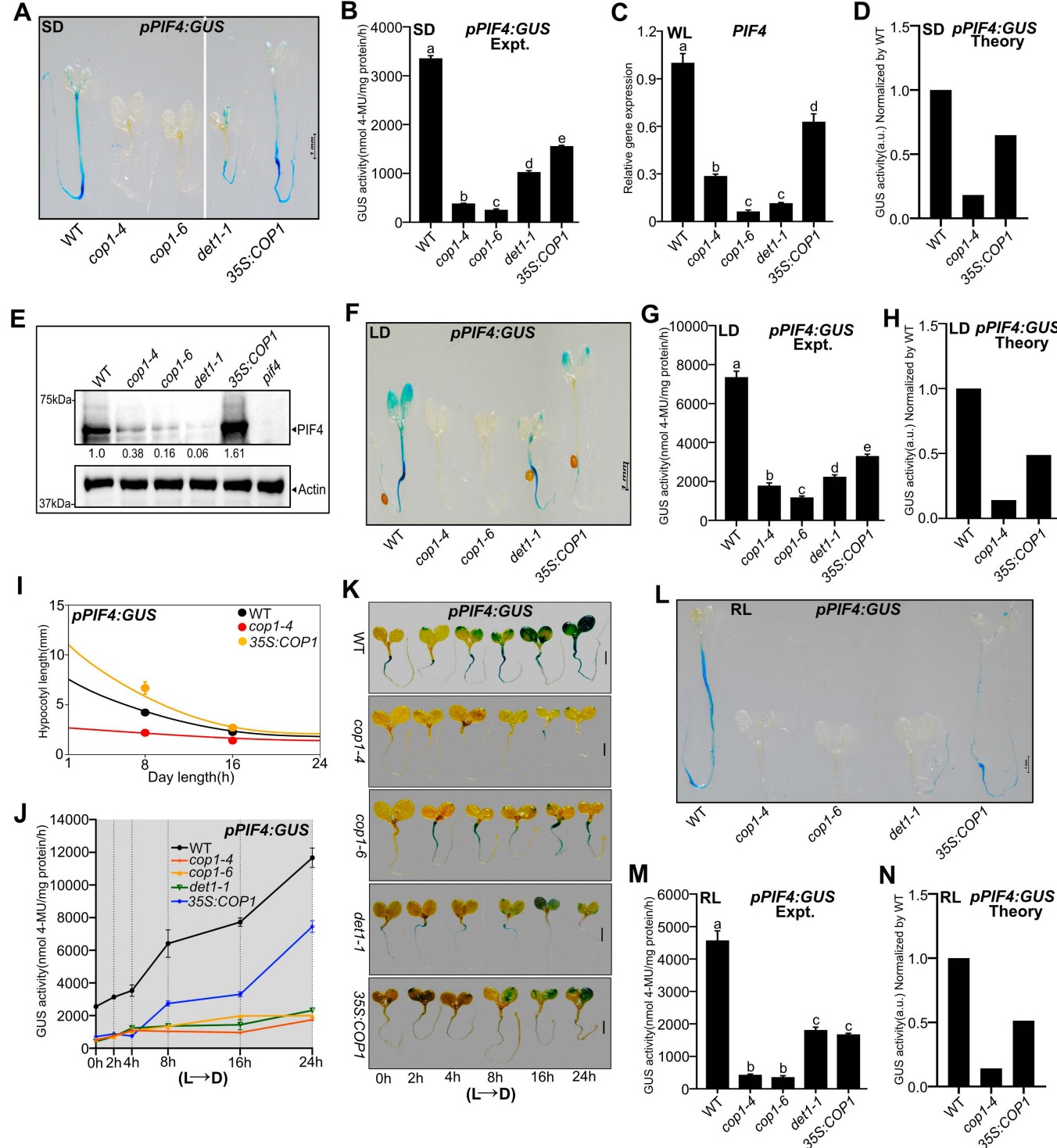

**Fig 3. COP1/DET1 module promotes PIF4 autoinhibition. (A and B)** The *pPIF4:GUS* stained images **(A)** and GUS activity **(B)** of six-day-old seedlings of indicated genotypes grown under SD in WL, and seedlings were harvested at ZT23. In A, the vertical line represents two separate images. **(C)** *PIF4* transcript level of indicated genotypes of six-day-old seedlings grown under SD in WL at ZT23. **(D)** Predicted GUS activity from our model, normalized to WT, under SD in WL. **(E)** Immunoblot analysis using PIF4 native antibody showing PIF4 protein levels compared to WT. Six-day-old seedlings were grown under SD in WL and harvested at ZT23. Values below the blots indicate the fold-change compared to WT. The *pif4* mutant was used as a negative control. **(F-H)** Images of GUS-stained seedlings **(F)**, measured GUS activity **(G)** at ZT4, and predicted GUS activity from the model **(H)** in

six-day-old seedlings grown under LD in WL. **(I)** Comparison between predicted and measured hypocotyl lengths of indicated genotypes. Filled circles represent experimental data (n = 20; Error bars represent standard deviations). **(J and K)** GUS activity **(J)** and GUS staining images **(K)** of constant light-grown seedlings (22°C) transferred to the dark for varying time intervals. The grey zone in 'J' represents the dark period. **(L-N)** GUS-stained images **(L)**, measured GUS activity data **(M)**, and predicted GUS activity from our model **(N)** of six-day-old seedlings grown under SD in RL (ZT23). Data are shown as mean ± SD. Different letters above the bar charts indicate a significant difference (One-way ANOVA with Tukey's HSD test, $P < 0.05$). The qRT-PCR data was obtained as fold-induction relative to WT at 22°C (Two biological replicates and three technical replicates). The experiment was repeated thrice, and similar results were obtained. See also S4 Fig.

*mutant*, it was comparable to WT (S4O Fig). Immunoblot data also suggest that in the *cop1–4* and *cop1–6* mutants, the CCA1 protein accumulation was higher compared to WT, while reduced in the *35S:COP1* line (S4P Fig). These results suggest that the reduced *PIF4* promoter activity in the *cop1* mutants is probably not due to reduced *CCA1/LHY* or *SHB1* transcript level or reduced CCA1 protein level.

We also performed western blot analysis to check the HY5 protein levels under SD photoperiod. We observed elevated protein accumulation in *cop1–4* and *cop1–6* mutants compared to WT (S4Q Fig). This observation aligns with previous reports that COP1 acts in the nucleus to repress light-responsive gene expression and developmental processes [69] and inhibits HY5 activity through protein degradation [70,71]. Moreover, there has been no report suggesting that HY5 can act upstream of PIF4, directly influencing its transcription. Nevertheless, HY5 has been shown to inhibit PIF4 targets [72]. Thus, HY5 likely has less influence on directly inhibiting *PIF4* gene expression, but it inhibits the expression of PIF4 targets. These results suggest that reduced *PIF4* promoter activity in the *cop1* mutants is likely independent of CCA1/LHY, SHB1 and HY5. However, the exact mechanism of how COP1 influences *PIF4* protomer activity is still largely unknown, and this will be investigated in the future.

### The phyB positively influences *PIF4* promoter activity and impedes autoinhibition

Phytochrome B is the major inhibitor of PIF4 function as it degrades PIF4 in response to RL, leading to exaggerated growth in the *phyb* mutant [19,22,23,59,60,73,74]. To understand the role of phyB on *PIF4* promoter activity, we introgressed *pPIF4:GUS* transgene into the *phyb-9* mutant and *35S:PHYB* overexpression lines. Data revealed that GUS staining and activity under SD at 22°C was significantly reduced in the *phyb-9* mutant but was significantly enhanced in the *35S:PHYB* (Fig 4A and 4B). Our mathematical model consistently predicted this experimental trend by assuming PIF4 autoinhibition (Fig 4C). In line with this, PIF4 protein was more stable in *phyb-9* but less stable in *35S:PHYB* than in WT (Fig. 4D). However, *PIF4* transcripts were higher both in *phyb-9* and *35S:PHYB* than in WT (Fig 4E). Thus, there is no linear one-to-one correspondence between the protein and transcript levels, again indicating underlying negative feedback. In *35S:PHYB,* there is much weaker negative feedback, resulting in higher *PIF4* promoter activity and likely producing more transcripts than the WT (Fig 4E). Meanwhile, in *phyb-9*, there is stronger negative feedback, leading to lower promoter activity, though producing mildly higher or comparable transcripts compared to the WT (Fig 4E).

Similar to the SD data, the GUS activity under LD was also lower in *phyb-9* but enhanced in *35S:PHYB,* compared to WT (Fig 4F and 4G). This trend also matched our model prediction (Fig 4H). Consistent with the *PIF4* promoter activity, our model predicted that the hypocotyl growth should be enhanced in *phyb-9* but reduced in *35S:PHYB,* compared to WT, for both SD and LD. This prediction was consistent with the measured hypocotyl length (Figs 4I, S5A and S5B). Like the SD data in WL, the predicted and measured *pPIF4:GUS* activity was significantly reduced in *phyb-9* but increased in *35S:PHYB* under RL (Fig 4J-4L). Also, similar to the WL, *PIF4* transcript in RL was more in both *phyb-9* and *35S:PHYB* than WT (Fig 4M), suggesting that *PIF4* autoinhibition may crucially depend on endogenous PIF4 concentration in each genotype. However, the GUS staining and activity of juvenile (S5C and S5D Fig) and adult plants (S5E-S5H Fig) showed lesser *PIF4* activity in *35S:PHYB* than WT, suggesting that phyB may have a diminished role in regulating *PIF4* activity beyond the seedling stage.

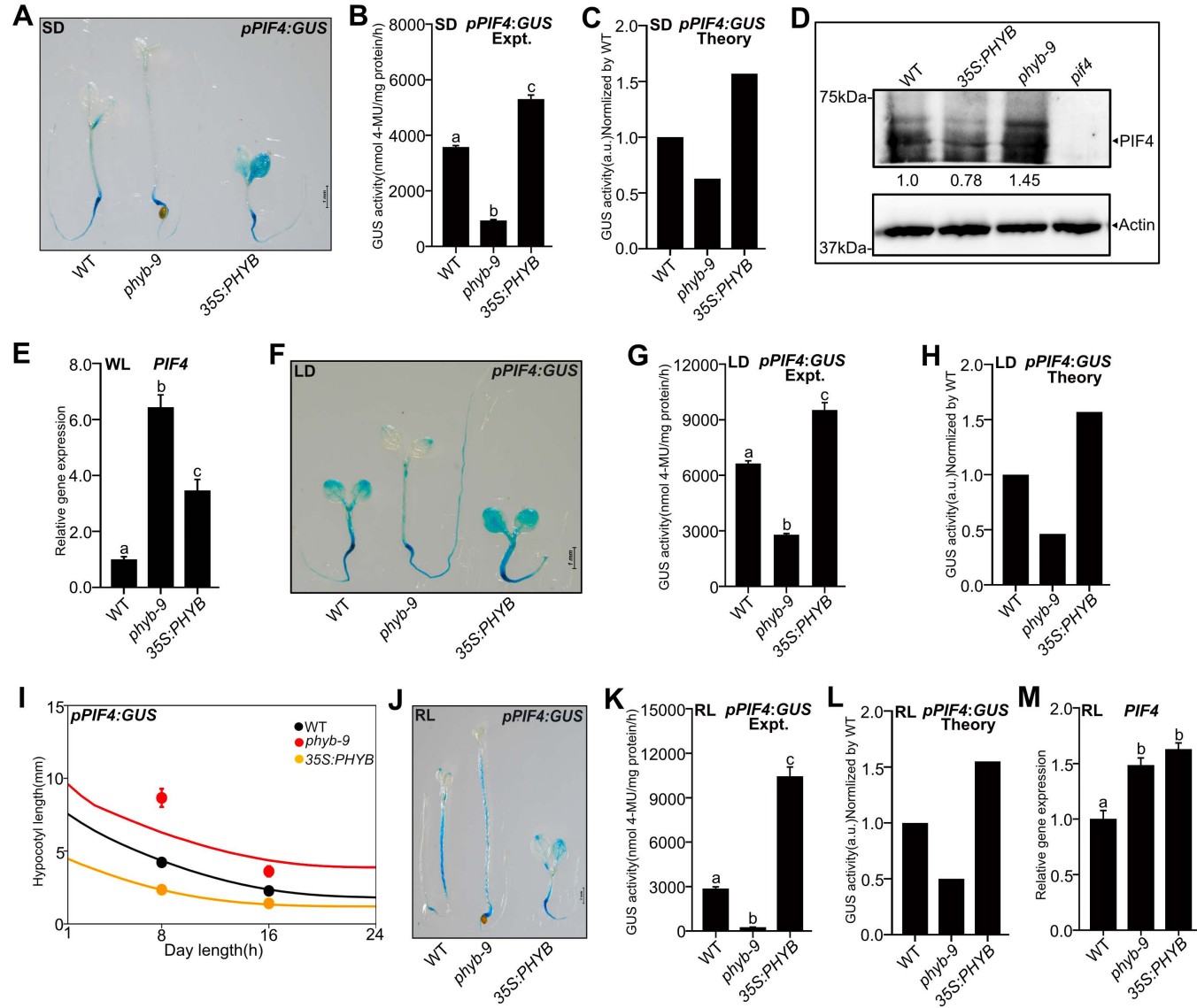

**Fig 4. The photoreceptor phyB negatively regulates PIF4 autoinhibition. (A-C)** Representative images of GUS staining **(A)**, measured GUS activity **(B)** and predicted GUS activity from the model **(C)** of six-day-old WT, *phyb-9*, and *35S:PHYB* seedlings grown under SD in WL. Both for GUS staining and activity measurements, seedlings were harvested at ZT23. **(D)** The PIF4 protein levels from six-day-old seedlings at ZT23 as revealed by immunoblotting (using anti-PIF4 antibody) in the respective genotypes grown under SD in WL. Values below the blots indicate fold-change compared to WT. Actin was used as the loading control. The *pif4* mutant was used as a negative control. **(E)** Real-time qPCR analysis of *PIF4* gene expression (at ZT23) from six-day-old WT, *phyb-9*, and *35S:PHYB* seedlings grown under SD in WL. **(F-H)** The *pPIF4:GUS* staining images **(F)**, GUS activity measurement **(G)**, and predicted GUS activity **(H)** from six-day-old seedlings grown under LD in WL. For GUS staining and activity measuremnts, seedlings were harvested at ZT4. **(I)** Predicted hypocotyl lengths for respective genotypes from the model. Filled circles denote experimental data (n = 20; Error bars represent standard deviations). **(J-L)** The representative GUS staining images **(J)**, quantitative GUS activity **(K)** at ZT23 and the model-predicted GUS activity **(L)** of six-day-old seedlings grown under SD in RL. **(M)** Fold change of *PIF4* transcript from six-day-old seedlings compared to WT under SD in RL (two biological and three technical replicates). Relative gene expression was calculated normalizing to the *EF1α*. Different letters in bar graphs indicate a significant difference (One-way ANOVA with Tukey's HSD test, P < 0.05). The experiment was repeated thrice, and similar results were obtained. See also S5 Fig.

## PIF4 differentially regulates *PIF4* promoter activity in a temperature-dependent manner

PIF4 is vital for thermosensory growth and reproductive transition in Arabidopsis [17,18]. Since *PIF4* gene expression is proportional to temperature [17,18,28,29], we compared the *pPIF4:GUS* staining and activities in WT, *pif4–101*, *PIF4-OE1* and *PIF4-OE2* seedlings grown at 22°C and 27°C under SD. As expected, the *PIF4* promoter activity was higher at 27°C compared to 22°C in the WT (Fig 5A and 5B). In the *pif4–101* mutant, the *pPIF4:GUS* activity was significantly higher than WT at 22°C, while it was reduced at 27°C (Fig 5A and 5B). In overexpression lines, the *PIF4* promoter activity was significantly lower than WT under both 22°C and 27°C, indicating the PIF4 autoinhibition (Fig 5A and 5B). Moreover, the *PIF4* promoter activity in *PIF4-OE1* and *PIF4-OE2* lines is slightly higher at 27°C than at 22°C. These observations could be due to higher PIF4 stability at 27°C than at 22°C and correspondingly increased PIF4 concentration threshold for autoinhibition. By including this hypothesis, our model could qualitatively reproduce the experimental trend of temperature dependence on *PIF4* promoter activity under SD (Fig 5C). Moreover, this hypothesis is consistent with the comparative PIF4 protein levels observed at 22°C and 27°C (Fig 5D). The tissue-specific GUS activity also showed a similar trend to the whole seedlings (Figs 5E, 5F, and S6A). Further, the activity under LD photoperiod showed a similar trend of temperature dependence in *pif4–101* and overexpression lines compared to the WT (S6B and S6C Fig). This trend was also reproduced in our model (S6D Fig). Next, we compared the GUS activities in hypocotyls, cotyledons, and roots. Similar to the whole seedling data, *pif4–101* showed reduced activity compared to WT at 27°C, and both overexpression lines showed a much stronger reduction in the GUS activity in all the tissues (S6E-S6G Fig).

Likewise, the temperature-dependent regulation of *PIF4* promoter activity became more apparent when we grew the seedlings at 22°C for six days in SD and exposed them to 27°C for different durations. Here, the *PIF4* promoter was induced over time in *pif4–101,* similar to the WT (Figs 5G and S6H). However, in overexpression lines, the promoter activity stayed several fold lower than WT (Figs 5G and S6H).

Since warmer temperature-mediated inhibition of phyB activity results in enhanced PIF4 activity [59,60], we checked *pPIF4:GUS* activity in *phyb-9* and *35S:PHYB* genotypes under both SD and LD. In the *phyb-9* mutant, the *pPIF4:GUS* activity was lower than WT at 22°C and 27°C (Figs 5H-J and S6I-K), similar to *PIF4* overexpression lines (Figs 5A-C and S6B-D). On the other hand, the *pPIF4:GUS* activity in *35S:PHYB* is higher than WT at 22°C and 27°C (Figs 5H, 5I, S6I and S6J), again suggesting that a higher phyB level leads to reduced autoinhibition. We further checked if our model could reproduce this experimental trend by assuming less phyB-mediated inhibition of PIF4 at higher temperatures. Our model successfully captured the effect of phyB on PIF4 autoinhibition in both temperatures under SD and LD (Figs 5J and S6K).

## PIF4 protein dynamics reveal differential autoinhibition of its promoter activity during day and night

Since PIF4 protein accumulation can vary over time and with temperatures, we monitored the PIF4 dynamics in the WT and overexpression lines (*PIF4-OE2*) at 22°C and 27°C under SD conditions. Our immunoblot analysis shows that the PIF4 protein accumulation peaks during the day in WT (around ZT2 at 27°C and ZT4 at 22°C), while the peaks appear slightly later in the overexpression lines (around ZT4 at 27°C and ZT8 at 22°C) (Fig 6A). In all cases, the PIF4 protein level drops slightly after the peak and is maintained throughout the night (Fig 6B and 6C). Also, as expected, the overall PIF4 protein level in *PIF4-OE2* is higher than that of the WT throughout the diurnal cycle. Moreover, *PIF4-OE2* shows a slightly higher peak in PIF4 protein compared to the WT. We also measured the *pPIF4:GUS* activity at the same time points at 22°C and 27°C under SD conditions. In the WT, the promoter activity peaks around ZT4 at 27°C and ZT8 at 22°C as reported [75], slightly later than the peak in PIF4 protein accumulation (Fig 6D and 6E). This suggests that the accumulation of proteins beyond a threshold dampens the promoter activity at the respective time points. Moreover, the promoter activity in *PIF4-OE2* is much lower than in WT throughout the day and night, suggesting strong autoinhibition since relative protein accumulation in these lines is maintained higher than in WT.

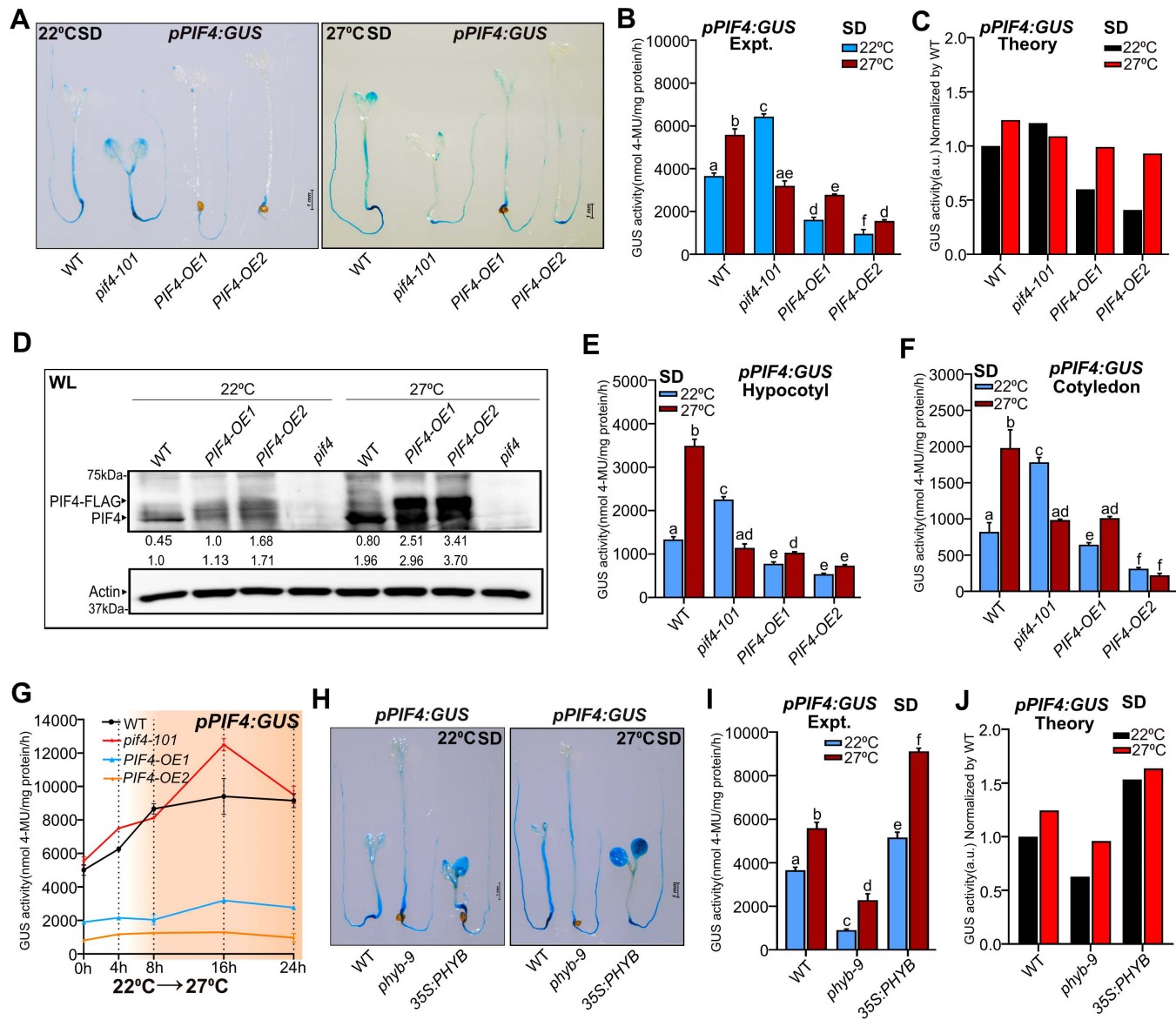

**Fig 5. PIF4 protein level regulates *PIF4* promoter activity in a temperature-dependent manner. (A and B)** Representative images of GUS-stained seedlings **(A)** and measured GUS activity **(B)** from six-day-old seedlings of respective genotypes grown under SD in WL at 22°C and 27°C, and the seedlings were harvested at ZT23. **(C)** Predicted GUS activity from the model at 22°C and 27°C for SD-grown seedlings. **(D)** PIF4 protein level from immunoblot (anti-PIF4 antibody) at ZT23 in indicated genotypes of six-day-old seedlings grown at 22°C and 27°C under SD. The arrow indicates the endogenous PIF4 and PIF4-FLAG bands. Values below the blots indicate fold-change compared to WT (22°C) to calculate endogenous PIF4 protein level (bottom). The values for the PIF4-FLAG band (top) were shown underneath each immunoblot, and normalization was done by setting *PIF4-OE1* (22°C) as 1. Actin was used as the loading control. The *pif4* mutant was used as a negative control. **(E and F)** Tissue-specific quantitative GUS measurement from six-day-old seedlings grown under SD at 22°C and 27°C at ZT23. The whole seedlings were cut into two sections, i.e., hypocotyl (E) and cotyledon **(F)**, and GUS activity was carried out individually (see Materials and Methods). **(G)** GUS activity of seedlings, grown at 22°C for five days and shifted to 27°C for one day. Tissue was collected from various time points. The yellow-shaded region represents warmer temperatures. **(H-J)** Representative GUS-stained images **(H)**, measured GUS activity **(I)** at ZT23, and model-predicted GUS activity **(J)** of six-day-old WT, *phyb-9*, and *35S:PHYB* seedlings grown at 22°C and 27°C in WL under SD. Different letters in bar charts indicate a significant difference (two-way ANOVA with Tukey's HSD test, P<0.05). The experiment was repeated three times with similar results. See also S6 Fig.

Since PIF4 protein accumulation and corresponding promoter activity are dynamic, the autoinhibition strength may vary over time. The promoter activity in the WT drops in the middle of the night and the beginning of the day, suggesting that autoinhibition could be higher in these time points compared to the mid-day (Fig 6D). Moreover, the promoter activity in the WT at 27°C is higher than at 22°C, suggesting lower autoinhibition strength at higher temperatures (Fig 6D). We thus assumed a similar variation of PIF4 autoinhibition strength in our mathematical model (Fig 6F). With this assumption, our model can qualitatively predict the similar dynamics of PIF4 protein and GUS activity for both WT and *PIF4-OE2* at 22°C and 27°C (Fig 6G and 6H). Together, our data reveal the complex interplay of PIF4 protein accumulation and the strength of negative feedback varying over time and with temperature.

## Elevated PIF4 dampens the expression of growth-promoting genes

PIF4-mediated promotion of thermosensory growth is linked to the direct activation of many genes involved in hormone biosynthesis, signalling, cell-wall elongation and floral transition [17,18,76]. We checked how increased PIF4 at high temperatures affects the expression of some key targets of PIF4, such as *YUCCA8 (YUC8), INDOLE-3-ACETIC ACID INDUCIBLE 29 (IAA29), XYLOGLUCAN ENDOTRANSGLYCOSYLASE 7 (XTR7), LONG HYPOCOTYL IN FAR RED (HFR1), and REPRESSOR OF GA (RGA) and HECTATE 1 (HEC1)*. In WT, *YUC8, IAA29* and *XTR7* expression was significantly upregulated at 27°C than 22°C (Figs 7A, 7B and S6L). However, compared to the WT, their expression in the *pif4–101* mutant was significantly lower at both 22°C and 27°C (Figs 7A, 7B and S6L). Therefore, PIF4 is essential to activate these growth-promoting genes. In *PIF4* overexpression lines, *YUC8, IAA19* and *XTR7* showed significant upregulation compared to the WT, irrespective of temperatures (Figs 7A, 7B and S6L). Notably, their expression was significantly reduced at 27°C compared to 22°C, suggesting that elevated PIF4 stability dampens the expression of these genes involved in growth.

Additionally, PIF4 directly activates *HFR1* [19,48], *HEC1* [50] and *RGA*, which form a negative autoregulatory loop. HFR1 and HEC1 (bHLH family members) and RGA (member of the DELLA family) physically interact with PIF4, resulting in either PIF4 transcriptional inhibition or PIF4 degradation [48,50,77]. Similar to *YUC8, IAA19* and *XTR7*, the expression of *HFR1, HEC1* and *RGA* was significantly upregulated in WT at 27°C than at 22°C, while their induction was compromised in the *pif4–101* mutant (S6M-S6O Fig). Their expression was also significantly elevated in the *PIF4-OE1* and *PIF4-OE2* lines compared to the WT at 22°C and 27°C (S6M-S6O Fig). Notably, unlike growth-promoting genes, *HFR1, HEC1* and *RGA* showed significantly enhanced expression in overexpression lines at 27°C than 22°C (S6M-S6O Fig), suggesting that elevated PIF4 function promotes the expression of these genes, likely to reinforce the negative autoregulation on PIF4.

## PIF4 threshold-dependent autoinhibition is key for optimal biomass and seed yield under varying photoperiod and temperature

Warm ambient temperature accelerates growth and development while compromising seed set and yield [7,78]. Though PIF4 is critical for warm temperature growth, its role in regulating biomass and seed yield remains elusive. To investigate this, we measured the rosette biomass (fresh weight) of three-week-old WT, *pif4–101* and *PIF4-OE* lines under varying photoperiods and temperatures. We found that rosette biomass was significantly reduced in the *pif4–101* than in WT under SD, while it accumulated more biomass than WT under LD (Fig 7C). However, increased PIF4 activity in *PIF4-OE1* and *PIF4-OE2* significantly reduced biomass compared to WT, independent of photoperiods (Fig 7C). Notably, the rosette biomass of *PIF4-OE2* was reduced considerably compared to *PIF4-OE1* under LD (Fig 7C). Also, the *pif4–101* mutant accumulated more biomass under 27°C SD than 22°C (Fig 7D). At 27°C, the biomass in *PIF4-OE1* was similar to WT, while it was significantly reduced in *PIF4-OE2* than *PIF4-OE1* (Fig 7D). Thus, the biomass depends on the concentration-dependent PIF4 autoinhibition in response to light and temperature.

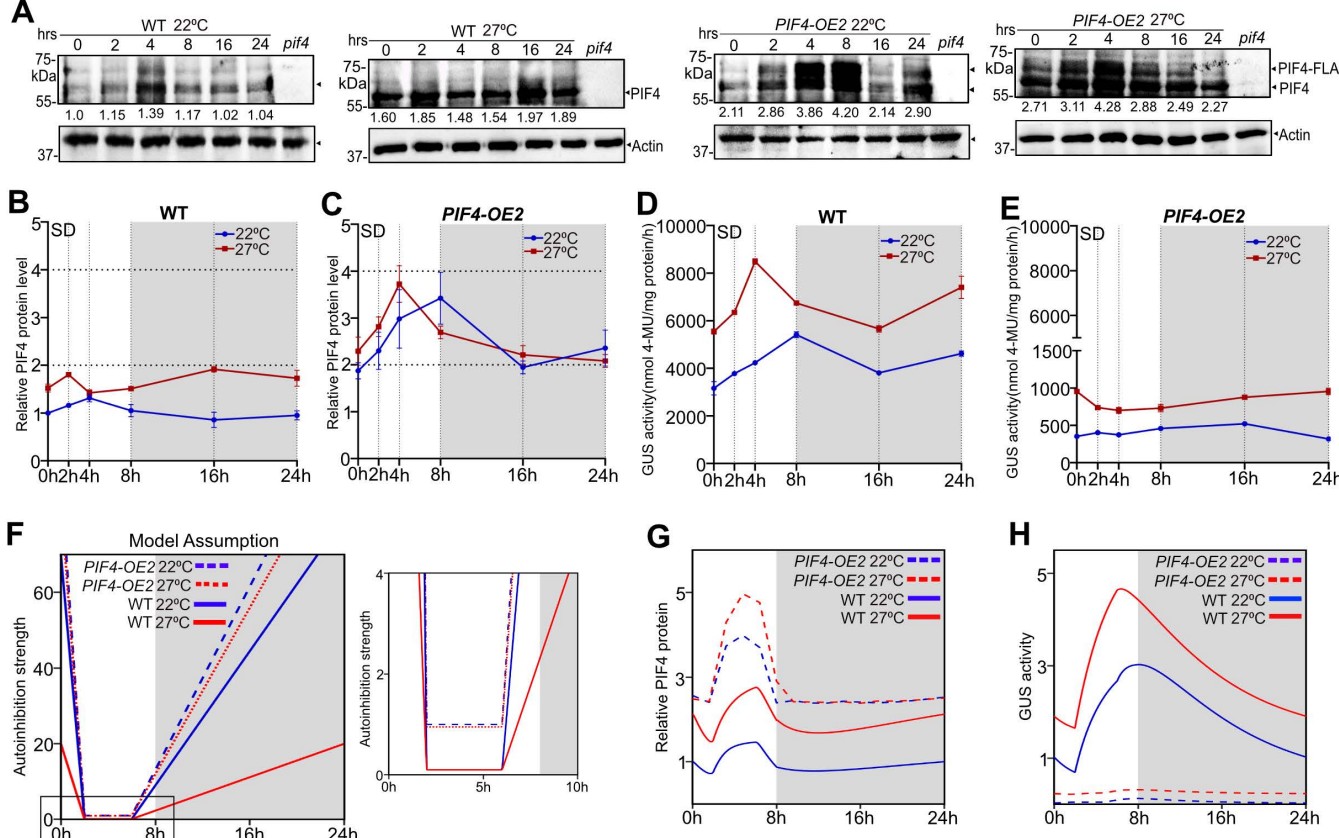

**Fig 6. Dynamics of PIF4 protein accumulation and autoinhibition in response to temperature. (A)** Immunoblot analysis at various time points of the PIF4 protein levels from seedlings of WT and *PIF4-OE2* grown at 22°C (up) and 27°C (down) under SD in WL. Tissue was harvested at ZT0 on the sixth day and grown for 24 hours. Arrowheads indicate PIF4 and PIF4-FLAG bands, which were considered for total PIF4 protein quantification. Values below the blots indicate fold-change compared to WT at 22°C, at ZT0. Actin was used as a loading control, and the *pif4* mutant as a negative control. All the immunoblotting experiments were performed at the same time, and identical conditions were maintained throughout the procedure. The experiment was repeated two times and similar results were seen. **(B and C)** The dynamics of relative protein levels of WT **(B)** and *PIF4-OE2* **(C)** at 22°C and 27°C. The relative protein levels were normalized by the WT value at 22°C, at ZT0. The data represents mean±SEM for two biological replicates. The dark period is shown in grey. **(D and E)** The measured GUS activities of WT **(D)** and *PIF4-OE2* **(E)** grown at 22°C and 27°C under the same conditions as described in 6.A. Data represents mean±SD for n ≥ 40 seedlings. The experiment was repeated three times with similar results. **(F)** The model assumption shows that the PIF4 autoinhibition strength varies over time. The autoinhibition strength is assumed to be higher at night and the beginning of the day, compared to the mid-day (left). Overall, the strength is higher in *PIF4-OE2* than in the WT. The autoinhibition strength at 27°C for WT is also assumed to be lesser than at 22°C, but this difference is negligible for *PIF4-OE2* (right). See S3 Table for parameter values. **(G and H)** Model predictions for PIF4 protein **(G)** and the GUS activity **(H)** over time at 22°C and 27°C. Data are normalized to respective WT values at 22°C, at ZT0. Grey shade denotes the dark period.

Similar to biomass accumulation, we also found that PIF4 is essential for optimal seed production. The WT produced significantly more seed yield under LD than SD, and *pif4–101* produced more seeds than WT under both SD and LD (Fig 7E). Conversely, overexpression lines produced significantly fewer seeds than WT, irrespective of the photoperiod (Fig 7E). Under warm temperatures, WT plants produced significantly fewer seeds than 22°C, while *pif4–101* produced more seeds than the WT at both 22°C and 27°C (Fig 7F). Interestingly, *PIF4-OE1* produced fewer seeds at 27°C than at 22°C, while *PIF4-OE2* did not produce any seeds specifically at 27°C (Fig 7F). This suggests that increased PIF4 levels, especially under warm temperatures, are detrimental to the optimal seed set.

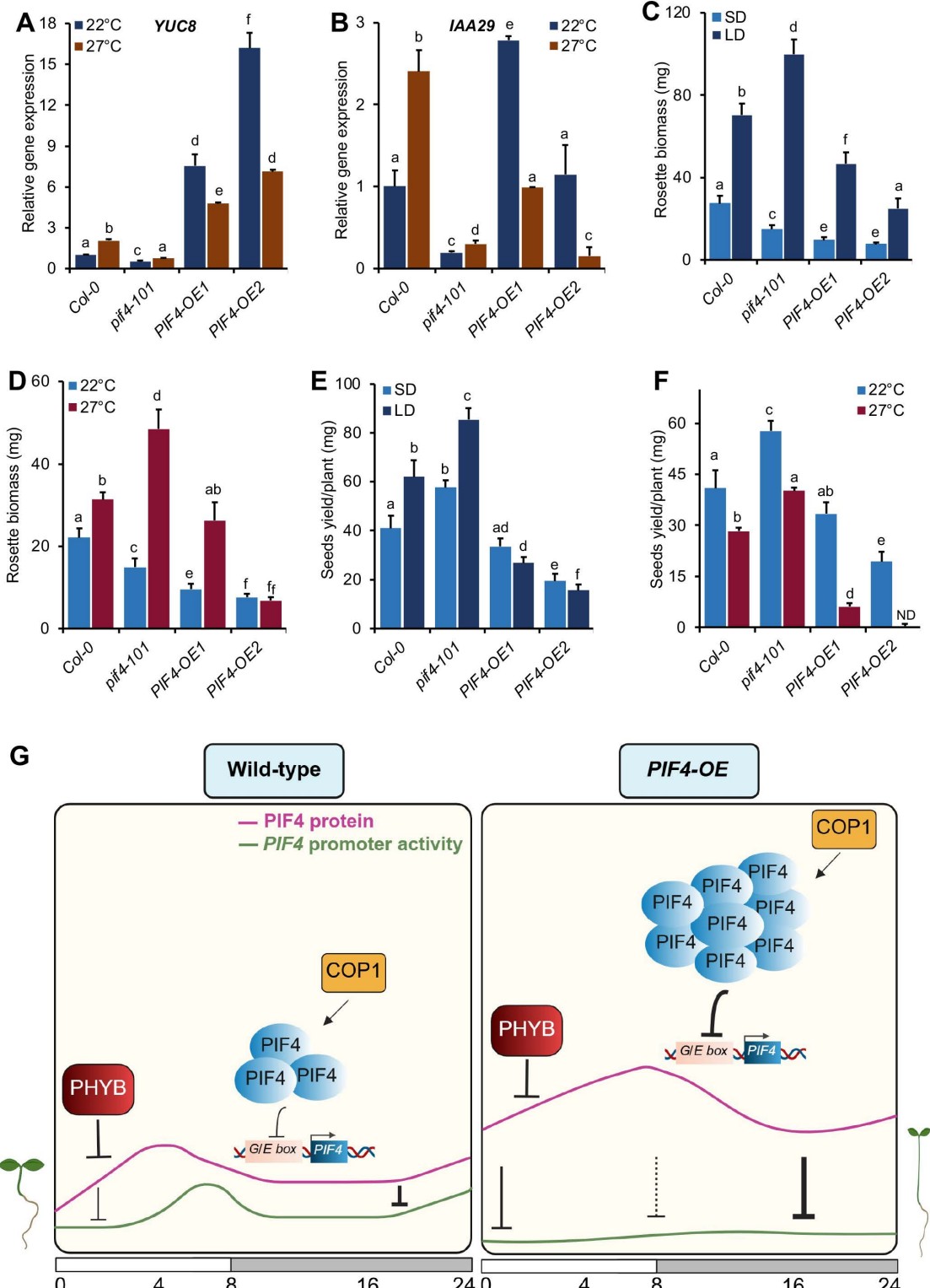

**Fig 7. Differential expression of growth-promoting and growth-inhibiting genes optimizes biomass production and seed yield. (A and B)** Expression of *YUC8* **(A)** and *IAA29* **(B)** genes at ZT23 as measured by RT-qPCR in respective genotypes of six-day-old seedlings grown under SD in WL at 22°C and 27°C. *YUC8* and *IAA29* are temperature-induced and growth-responsive genes. Relative gene expression was calculated normalizing

to the *EF1α*. **(C and D)** Rosette biomass of three-week-old plants WT, *pif4-101, PIF4-OE1* and *PIF4-OE2* lines grown under different photoperiods at 22°C **(C)** and at different temperatures under SD **(D)**. Data represent mean±SD (n ≥ 6). **(E and F)** Seed yield of indicated genotypes grown in different photoperiods at 22°C **(E)** and at different temperatures under SD **(F)**. Data represent mean±SD (n ≥ 10 plants). Different letters in bar charts indicate a significant difference (two-way ANOVA with Tukey's HSD test, P < 0.05). The experiment was repeated three times with similar results. **(G)** The graphical summary shows the dynamic nature of PIF4 protein accumulation and the corresponding autoinhibition of its promoter activity in WT and *PIF4* overexpression backgrounds. In the WT, mild autoinhibition during the day and stronger inhibition at night. In the overexpression background, autoinhibition is stronger overall than in WT. Created in BioRender. Das, S. (2025) https://BioRender.com/zxysze4. See also - S6 and S7 Figs.

## Discussion

PIF4, the key integrator of multiple environmental cues such as light and temperature, is strictly regulated to control the signalling output in tune with the endogenous hormones/metabolic fluxes [10,12,21]. Here, we demonstrate a novel mechanism through which PIF4 negatively autoregulates its promoter activity to optimise growth and grain yield in response to varying photoperiods and temperatures. Consistently, the *pif4* mutant showed higher promoter activity than the WT, while the *PIF4* overexpression lines displayed strongly reduced activity under both white and monochromatic red light (Fig 1A-D and 1M-P). On the other hand, the *COP1* overexpression also showed lower promoter activity than the WT, probably due to higher *PIF4* levels enforcing autoinhibition (Fig 3A-H). Moreover, overexpression of phyB, which inhibits PIF4 stability, phenocopied the *pif4* mutant, showing higher promoter activity (Fig 4A-H). Similarly, *phyb-9* mutant behaves like the *PIF4* overexpression lines in repressing *PIF4* promoter activity (Fig 4A-4H). Together, these data provide evidence that PIF4 autoinhibits its transcript accumulation.

A crucial insight from our study is that we cannot always expect a linear one-to-one correspondence between the PIF4 protein and its transcript levels in the presence of negative feedback. For instance, *35S:PHYB* overexpression has reduced PIF4 protein levels and hence weaker autoinhibition, leading to higher promoter activities and transcript accumulation than WT. On the other hand, *PIF4-OE2* has significantly more PIF4 protein and stronger autoinhibition, resulting in lower promoter activities and lower or similar transcript levels compared to WT. However, when the PIF4 protein levels are mildly higher than WT, as in the cases of *PIF4-OE1* and *phyb-9* mutant, the correspondence between the PIF4 proteins and transcript becomes nonlinear due to negative feedback. Although we may expect lower transcripts due to higher PIF4 stability, a mild increase in PIF4 may not be enough to cross the threshold needed to lower the transcript level significantly. Also, since the autoinhibition is a dynamic process, the transcript levels for *PIF4-OE1* and *phyb-9* mutant may become temporarily higher than or comparable to the WT, highlighting the complexity of such regulation.

Notably, we observed differential promoter activity in two *PIF4* overexpression lines with distinct PIF4 protein levels, suggesting that PIF4 autoinhibition depends on endogenous PIF4 concentration (Fig 1A-1G). Furthermore, *cop1* mutants showed negligible to weak *PIF4* promoter activity compared to WT, suggesting that the endogenous PIF4 level in these genotypes is very low (Fig 3A-3H). Similarly, the *pif4* mutant at higher temperatures showed reduced *PIF4* promoter activity than WT, though the activity at lower temperatures is much higher than WT (Figs 5A-5C and S6B-S6D). This differential temperature response may indicate that *PIF4* autoinhibition takes place when PIF4 concentration reaches above a threshold. In line with this, a recent report suggests that PIF4 may even autoactivate below the threshold concentration [79]. In general, the PIF4 threshold levels in cells may change depending on light and temperature, which further dictates whether the *PIF4* transcription should be activated or repressed.

We further developed a minimal network model using the hypothesis of a concentration threshold-dependent PIF4 autoinhibition, including the main interactions between PIF4, COP1, ELF3, and phyB (Fig 2A). This model successfully predicted the measured hypocotyl lengths in various genetic backgrounds and also recapitulated the observed trends of *PIF4* promoter activity. This further supports our claim that PIF4 inhibits itself in a concentration-dependent manner. Interestingly, our model showed unbounded high hypocotyl growth without PIF4 autoregulation, emphasizing its critical role in

controlling plant growth (Fig 2B). We also explored how this mechanism relates to the expression of PIF4 targets affecting growth and overall fitness, such as rosette biomass and seed yield in varying photoperiods and temperatures (Fig 7A-7F).

In summary, higher endogenous PIF4 levels promote lesser *PIF4* promoter activity and correlate with longer hypocotyl (Fig 7G). Moreover, the PIF4 protein accumulation and subsequent autoinhibition of its promoter activity are complex and dynamic (Fig 6). In WT, our data and model suggest that the autoinhibition strength is higher at night, during which PIF4 protein accumulates and passes a threshold to inhibit its own transcription. This negative feedback leads to asynchronous oscillation of PIF4 protein and corresponding promoter activity for effective maintenance of the PIF4 protein level during the night (Fig 7G). However, in overexpression lines, the relative PIF4 protein is much higher than in WT, resulting in a stronger autoinhibition. Moreover, warmer temperatures also lead to stronger autoinhibition, likely due to reduced phyB activity and increased PIF4 accumulation (S7 Fig). Together, we propose that PIF4 undergoes autoinhibition, either regulating its own transcription in a concentration-dependent manner or via promoting downstream regulators that inhibit PIF4.

## Materials and methods

### Plant materials and growth conditions

Unless otherwise specified, all genetic materials used in this study are in the Col-0 background. The various genetic resources, *pPIF4:PIF4-FLAG (PIF4-OE1* and *PIF4-OE2* [20]), *35S:PIF4-HA (PIF4-OE3*, [15]), *pPIF4:GUS* (this study), *pPIF4:LUC* [80], *pif4–101* [19], *cop1–4* [81], *cop1–6* [81], *det1–1* (NASC, N6158), *35S:COP1* [67], *phyb-9* [82], *35S:PHYB-GFP phyb-9* [83], *hy5–215* [84], *cca1–1* (N67781) were used. For various experiments in this study, seeds were surface-sterilized (70% ethanol + 0.05% Triton X-100) and germinated on Murashige and Skoog (MS; pH: 5.7) plates containing 1% sucrose (Sigma-Aldrich) and 0.8% w/v agar (Himedia Laboratories, India) following stratification for three days at 4°C in the dark. Next, seeds were transferred to 22°C under LD photoperiod (LD; 16h light/8 h dark) in the plant growth chamber for germination. Upon germination, seedlings were either transferred to 27°C or retained at 22°C for six days unless otherwise specified. The light intensity of 100 μmol m$^{-2}$ s$^{-1}$ was used to grow seedlings for all the experiments. Experiments were performed under dark or short-day (SD; 8h light/16 h dark), long-day (LD; 16h light/8 h dark), or constant light (24 h light) conditions as specified. For monochromatic light experiments, red light (RL) was used at 60 μmol m$^{-2}$ s$^{-1}$ intensity. Experiments related to monochromatic lights were performed under short-day conditions.

### Vector construction and generation of transgenic lines

The promoter-reporter line *pPIF4:GUS* was constructed by gateway-based cloning. Genomic DNA from Col-0 was used to amplify the *PIF4* (1236 bp) promoter and the PCR fragment was cloned into pENTR/D-TOPO and inserted into the pGWB633 (carboxy-terminal GUS) vector. All recombinant clones were validated through restriction analysis and were sequence-confirmed.Then, the specific constructs were transformed into *Agrobacterium tumefaciens GV3101*. Transgenic plants were obtained by transforming Col-0 using the floral-dip method. Transgenics were selected on MS+BASTA (7.5 μg/μl) antibiotic. Two independent homozygous transgenic lines (Lines #1 and #2) were analysed for GUS activity, and transgenic line #1 was used for further analysis.

### Generation of *pPIF4:GUS* in various mutant and overexpression lines

For better understanding, we labelled PIF4-overexpressor lines and categorised them based on their hypocotyl length; *pPIF4:PIF4-FLAG-OE1 (PIF4-OE1)* has a moderately long hypocotyl phenotype while *pPIF4:PIF4-FLAG-OE2 (PIF4-OE2)* has very long hypocotyls and *35S:PIF4-HA (PIF4-OE3)* has significantly long hypocotyls. For generating *pPIF4:GUS* promoter-reporter transgene in various genotypic backgrounds, we crossed *pPIF4:GUS transgenic line* separately with *pif–101, PIF4-OE1, PIF4-OE2, PIF4-OE3, cop1–4, cop1–6, det1–1, 35S:COP1, phyb-9, 35S:PHYB-GFP*, to obtain F$_1$

seeds. $F_1$ seeds were further propagated to get $F_2$. Next, $F_2$ seeds were screened by plating on MS along with appropriate antibiotic selection. Desired seedling combinations were selected based on antibiotic selection and hypocotyl phenotype. The selected plants were further genotype and/or phenotype confirmed in the adult stage, along with GUS staining for the *pPIF4:GUS* transgene (Please refer to S4 Table for primers used for genotyping). For generating *pPIF4:LUC* promoter-reporter transgene in the background of *pif4–101* and *PIF4-OE2*, we crossed with the lines separately, and $F_2$ screening was performed with appropriate antibiotic selection and homozygous lines were selected. The putative homozygous lines were further confirmed in the next generation by antibiotic selection and phenotypic analysis (hypocotyl measurement) before further analysis.

### Hypocotyl measurements

Six-day-old seedlings of various genotypes grown under either SD or LD were used to measure hypocotyl lengths. On the sixth day, ~20–25 seedlings per genotype were aligned on an Agar plate containing charcoal before being photographed along with a scale. Later, hypocotyl lengths were measured using NIH ImageJ software (https://imagej.nih.gov/ij).

### GUS histochemical staining assay

For the GUS histochemical assay, six-day-old seedlings, three-week-old juvenile or six-week-old adult transgenic plants grown under specified growth conditions were used. Histochemical assay for *β-Glucuronidase (GUS)* was performed in the seedlings (whole seedlings, different tissues such as hypocotyl, cotyledon and roots), juvenile and adult plants (rosette leaf and stem). GUS histochemical assay was performed as described [85] with minor modifications. Approximately 20–25 seedlings were used for histochemical staining. Tissue from the control and transgenic plants was harvested and fixed in fixation buffer (2% formaldehyde, 50 mM sodium phosphate (pH 7.0), 0.05% Triton X-100), and vacuum infiltrated for 4–5 min on ice and kept at room temperature for 10 min. The fixation buffer was removed, and the material was washed twice with 50 mM sodium phosphate buffer (pH 7.0) to remove the fixative buffer. The tissue samples were stained using staining buffer [(1.5 mM of X-gluc, 50 mM sodium phosphate (pH 7.0) and 0.1% Triton X-100)] by vacuum infiltrating for 5–10 min and then wrapped with aluminium foil and incubated at 37°C overnight in the dark. The next day, samples were observed for staining patterns. Tissue was destained extensively with a destaining solution (Ethanol:Acetic acid: Glycerol at a ratio of 4:4:2) for 10 minutes, and the destained seedlings were stored in 10% glycerol. In a few cases (constant light-to-dark transition experiments), we also used 70% ethanol after overnight staining to remove the chlorophyll for better visualisation of GUS staining. Representative seedlings from different genotypes and growth conditions were aligned on suitable media and photographed.

### GUS spectrometric assay

Approximately 40–50 seedlings from ~100 mg fresh tissues (seedlings/whole plants/ parts of the plants) were harvested in a micro-centrifuge tube frozen in liquid nitrogen and ground in 100µl of extraction buffer [50 mM sodium phosphate (pH 7.0), 5 mM DTT, 1 mM EDTA, 0.1% sarcosyl, 0.1% Triton X100] at 4°C. The sample was transferred into a fresh microcentrifuge tube and centrifuged at 10,000 rpm for 5 min at 4°C. Supernatant was separated and moved to a the fresh tube. 10 µl of protein extract was added to the 190 µl of assay buffer (1 mM MUG in extraction buffer) and incubated at 37°C for 15 min. Next, 180 µl of 0.2 N $Na_2CO_3$ stop buffer was added to a 20 µl reaction mix to stop the reaction. GUS activity was determined by the fluorometric assay as described [85]. Total protein was quantified in the extract using the Bradford assay. *β-Glucuronidase (GUS)* specific activity was recorded as nanomoles of 4-MU formed per milligram of protein per hour (nanomoles of 4-MU/mg protein/h) formed per milligram of from the initial velocity of the reaction. Finally, the GUS activity was calculated by comparing the spectrophotometric reading to the MU standard and normalizing it to the total protein content and dilution factor, wherever applicable.

## Sample collection for tissue-specific GUS activity measurement

Tissue-specific GUS spectrometric assay was carried out from six-day-old seedlings grown in square plates. Hypocotyl, cotyledon and roots were excised using a scalpel blade and collected into Eppendorf tubes before being frozen in liquid nitrogen. Total protein was extracted, and a GUS spectrometric assay was carried out.

## Luciferase activity assay

We detected luciferase (LUC) activity using Firefly Luciferase Assay Kit 2.0 (Biotium). Approximately 25–30 fresh seedlings (one-week-old) were harvested in a microcentrifuge tube and snap chilled in liquid nitrogen and ground in 150 µl of extraction buffer [50 mM Tris HCl pH 8.0, 150 mM NaCl, 10% glycerol, 5mM DTT, 1% (v/v) Protease Inhibitor Cocktail, 1% NP40, 0.5mM PMSF]. The lysate was cleared by centrifugation at 10,000 rpm for 10 min at 4°C and transferred into a new tube. The tubes were placed at 4°C until ready to assay. Firefly working solution was prepared by adding D-Luciferin (10 mg/ml) to assay buffer at a ratio of 1:50. 20 µl of total protein extract was directly added to the plate (Cat. No 781665, Brand 96 well plate). The 100 µl of firefly working solution was added to the protein sample and mixed by gentle pipetting. Then, the plate was immediately placed in a plate reader, and the firefly luminescence measurement was recorded and represented as the average of counts/s$^{-1}$ (cps) per well. Total protein was quantified in the extract using the Bradford assay, and the level of luciferase was calculated by normalizing it to the total protein content. The dark condition was maintained during the experiment.

## RNA extraction and gene expression analysis by RT-qPCR

For gene expression analysis using quantitative-PCR (qPCR), RNA was extracted using RNeasy Plant mini kit (QIAGEN) with on-column DNase I digestion according to the manufacturer's instructions. RNA was quantified using NanoDrop, and approximately 2.0 µg of total RNA was converted into cDNA using a Verso cDNA synthesis kit (Thermo-Fisher scientific) and oligo dT according to the manufacturer's instructions. Exactly 2.0 µL of 1:20 diluted cDNA was used for qPCR using a 2x SYBR Green Master Mix kit. qPCR experiments were performed in a QuantStudio 5 Real-Time PCR System (Applied Biosystems). *EF1α* (AT5G60390) was used as an internal control for normalization. Details of the oligonucleotide primers used are provided in S4 Table.

## Protein extraction, SDS PAGE running and western blot analysis

Approximately 100 mg of tissue was harvested in a microcentrifuge tube, snap frozen in liquid nitrogen and ground in 200 µl of protein extraction buffer [50 mM Tris HCl pH 8.0, 150 mM NaCl, 10% glycerol, 5mM DTT, 1% (v/v) Protease Inhibitor Cocktail, 1% NP40, 0.5mM PMSF]. The protein extract was centrifuged at 10,000 rpm for 15 min at 4°C to pellet down the debris. The supernatant was then transferred to a fresh tube and maintained at ice-cold conditions. An aliquot of 3–5 µl was removed in a separate tube to estimate protein concentration by Bradford assay. The protein samples diluted to 1µg/µl were boiled for 10 min at 70°C and 50 µg denatured protein samples were loaded onto the SDS PAGE (10% gel)for separating proteins and run for approximately 4 hr. Separated proteins were transferred to the PVDF membrane at 90 volts for 1 hr in transfer buffer (Tris 48 mM, Glycine 39 mM, 20% methanol pH 9.2) in the wet transfer method in the cold room. The membrane was stained with Ponceau-S to confirm the protein transfer and then washed with sterile MQ water. The membrane was then incubated on a rotary shaker for 2 hr in 10 ml blocking buffer (5% non-fat dry milk in Tris-buffered saline (TBS) with 0.05% Tween-20) at room temperature. The blocking reagent was removed, and the affinity-purified primary antibody diluted (1:2500–1:10,000) in 10 ml TBS with 0.05% Tween-20 was added and incubated overnight with shaking in the cold room. The next day, the membrane was washed thrice with 10 ml of wash buffer (TBS and 0.05% Tween-20) for 5 min each. The secondary antibody conjugated with HRP diluted (1:5,000–10,000) in 10 ml TBS with 0.05% Tween-20 was added and incubated for 2 hr with shaking at room temperature. The membrane was

washed thrice with 10 ml of wash buffer at room temperature. The blot was developed using the Super Signal West Femto chemiluminescent substrate kit (Pierce) and following the instructions provided by the manufacturer. A substrate working solution was prepared by mixing peroxide and Luminol/enhancer solutions in a 1:1 ratio. The blot was incubated in that working solution for 5 min in the dark. The blot was removed from the working solution and observed in Chemi-Doc by exposure for different time durations depending on signal strength. The bands were quantified by ImageJ software. The total PIF4 protein (PIF4 endogenous + PIF4-FLAG) is quantified by considering two bands and normalized to WT (by setting WT to 1). The PIF4-FLAG band was also quantified separately by normalizing to *PIF4-OE1* (by setting *PIF4-OE1*–1). Actin was used as a loading control, and *pif4–101* was used as a negative control in all the immunoblot experiments. For all the western blots, an identical condition was maintained. The antibody used for detecting endogenous PIF4 level was PIF4 (goat antibody, Agrisera, Cat no. AS163955 at dilution 1:2500). The Actin antibody (abcam, Cat no. ab197345 at dilution 1:10000) was used as the loading control. The anti-CCA1 (Agrisera, Cat no. AS13 2659), anti-COP1 (Agrisera, Cat no. AS20 4399), and anti-HY5 (Agrisera, Cat no. AS12 1867) were used for this study.

### Yeast-one hybrid assay

To generate constructs for yeast one-hybrid assays, bait and prey were prepared. CDS of *PIF4* (encoding full-length protein) from Col-0 cDNA and full-length promoter of *PIF4* (containing G/E-boxes and PBE-box) from genomic DNA of Col-0 were amplified. CDS of *PIF4* was cloned into the pGADT7 vector. The PCR fragment of the promoter of *PIF4* was cloned into the *pAbAi* vector, respectively, containing Hind III + Kpn I restriction cut sites by restriction-based protocol. The positive colonies were confirmed by colony PCR with gene-specific primers and sequencing by using vector-specific primers. The constructs were transformed into yeast strains *YM4271* and *Yα1867,* and yeast-one hybrid screening was performed in SD/-Ura/-Leu containing 150–200 ng/ml Aureobasicidin A for around seven days at 30°C.

### Chromatin Immunoprecipitation (ChIP) assay

ChIP was carried out as described [18] with minor modifications. For this experiment, *PIF4-OE2* seedlings were grown on MS medium for seven days; they were maintained at 22°C under short days (8 h light/16 h dark) in white light. Seedlings (approximately 2.5 gm) were harvested in dim light and directly cross-linked with 1% formaldehyde. ChIP was done using paramagnetic Dynabeads coated with monoclonal anti-FLAG-HRP conjugated antibody (Abcam, ab49763) following the manufacturer's instructions. Beads were washed four times with the immunoprecipitation buffer and two washes with Tris-EDTA buffer (TE). Reverse cross-linking was done by boiling at 95°C for 10 min in the presence of 10% Chelex (BioRad), followed by proteinase K treatment at 50°C. The qPCR was performed using SYBR Green dye in QuantStudio 5 Real-Time PCR System (ABI), and enrichment was calculated relative to wild-type controls. PIF4 binding to its promoter was performed using a set of primers spanning the promoter regions covering either PBE-box, E-box or G-box elements. Oligonucleotide sequence details are provided in S4 Table.

### Sampling of biological samples for various assays

As indicated in the respective sections, all experiments were performed with at least two biological replicates. Unless otherwise mentioned, at least 20 seedlings (six-day-old) were used for measuring hypocotyl length. ZT23 for SD and ZT4 for LD were followed for harvesting tissue samples for various experiments unless otherwise specified. For the time-course experiment, the seedlings were grown for five days under SD. Tissue was harvested from the sixth day at ZT0 until ZT24 for various durations under 22°C and 27°C. For immunoblot experiments, six-day-old seedlings were used, and the experiment was repeated thrice and similar results were seen. For GUS staining, ~20–25 seedlings were harvested and stained. The stained representative seedlings were used for the reference picture. For GUS activity measurement, tissue was harvested in two biological replicates, and three technical replicates were used. For gene expression analysis, the seedlings were grown for six days, tissue was harvested, and three independent biological replicates were sampled at

specific time points. The *EF1αI* gene was used as a housekeeping control for RT-qPCR studies. For the rosette biomass and seed yield experiments, at least six biological replicates were used, respectively.

## Mathematical model

Our model is adapted from a published model [68] that we extended by introducing the proposed autoinhibition by PIF4. In our model, $B(t)$ denotes the photoactivated form of phyB; $E(t)$ and $C(t)$ denote the ELF3 and COP1, respectively; and $P(t)$ denotes the concentration of PIF4. Also, $B(t)$, $E(t)$, $C(t)$, and $P(t)$ represent the cellular concentrations of respective proteins at a time, $t$. Moreover, $G(t)$ denotes the hypocotyl length. Here, $G(t)$ is a coarse-grained variable that represents the collective effect of all growth-promoting genes targeted by PIF4. Additionally, $F(t)$ denotes the GUS concentration, representing the transgenic promoter activity (i.e., the GUS activity). Note that all molecular concentrations are expressed in arbitrary units, while the hypocotyl length is in *mm*. The following set of ordinary differential equations (ODEs) describes the temporal dynamics of the variables.

$$\frac{dB(t)}{dt} = p_B(T)L(t)\left(mut_B - B(t)\right) - k_r(T)B(t)$$

$$\frac{dE(t)}{dt} = p_E(t, T, D, mut_E) - d_{EC}C(t)E(t) - d_E E(t)$$

$$\frac{dC(t)}{dt} = mut_C\left[p_{CL}(T)L(t) + p_{CD}\left(1 - L(t)\right)\right] - d_C C(t)$$

$$\frac{dP(t)}{dt} = mut_P \frac{p_P}{1 + p_{PE}(T)E(t) + p_{self}P(t)} - \frac{d_P P(t)}{1 + k_{PC}C(t)} - d_{PB}B(t)P(t), \quad P > P^*$$

$$= mut_P \frac{p_P}{1 + p_{PE}E(t)} - \frac{d_P P(t)}{1 + k_{PC}C(t)} - d_{PB}B(t)P(t), \quad P < P^*$$

$$\frac{dF(t)}{dt} = k_0(T) + \frac{p_F}{1 + p_{FP}P(t)} - d_F F(t), \quad P > P^*$$

$$= k_0(T) - d_F F(t), \quad P < P^*$$

$$\frac{dG(t)}{dt} = p_G + k_G \frac{p_{GP}P(t)}{1 + p_{GE}E(t) + p_{GB}B(t)}$$

Here, $L(t)$ is a binary variable denoting the presence or absence of light, which is either 0 in dark or 1 in white/red light. In the ODEs (Equation 1), decay rates are, in general, denoted by $d_K$ where $K$ symbolically represents the variable $K$ (see the detailed parameter descriptions in S1 and S2 Tables). Similarly, $p_K$, in general, denotes the production rate of the variable $K$. Specifically, the production rate of ELF3 (denoted by $p_E$) is modelled by an oscillatory function (see Equation 2). The production rates of COP1 are denoted by $p_{CL}$ and $p_{CD}$, in light and dark, respectively. Notably, we included

a multiplicative factor $mut_K$ that modifies the production rate of variable $K$ in the corresponding knock-out and over-expressor lines, i.e., $mut_K = 1$ for the wild type, $mut_K < 1$ for the mutants, and $mut_K > 1$ for overexpresson lines (see S2 Table for specific values of these multipliers in different genotypes). In general, the parameter, $p_{MN}$, denotes the interaction strength between variables $M$ and $N$ (see the detailed descriptions in S1 and S2 Tables).

Moreover, we assumed that PIF4 autoinhibition occurs above a threshold PIF4 concentration denoted by $P^*$. Consequently, PIF4-induced inhibition of GUS also takes place when PIF4 concentration is above $P^*$. Note that the threshold PIF4 concentration ($P^*$) depends on a specific genotype, in general. The intensity or strength of autoinhibition for PIF4 synthesis is denoted by $p_{self}$. Furthermore, $k_0$ is the basal production rates of GUS. To model the hypocotyl growth, we assumed that $k_G$ is a coarse-grained parameter representing the conversion of PIF4 target gene expressions into the overall growth. Finally, some parameters are assumed to depend on the temperature, $T$, and have different values at $22^oC$ and $27^oC$ (see S1 and S2 Tables).

Previous studies [68] showed that ELF3 has an oscillatory nature depending on the diurnal days. Thus, we incorporated the ELF3 synthesis by an oscillatory function as below:

$$p_E(t, T) = mut_E\, p_{E1}(T) + p_{E2}(T), \quad if\ D = 0\ hour$$

$$= mut_E\, p_{E1}(T) - p_{E2}(T)\left(-1 + \frac{2}{1 + \exp(-\alpha t_0)} - \frac{2}{1 + \exp(-\alpha t_1)} + \frac{2}{1 + \exp(-\alpha t_2)}\right), \quad if\ 0 < D < 24\ hour$$

$$= mut_E\, p_{E1}(T) - p_{E2}(T), \quad if\ D = 24\ hour \tag{2}$$

Here, D represents the time of light exposure (i.e., the day length). The production rate of ELF3 oscillates between $(p_{E1} + p_{E2})$ and $(p_{E1} - p_{E2})$. Also, $t_0 = mod(t, 24)$, $t_1 = t_0 - D$, $t_2 = t_0 - 24$. The parameter $\alpha$ defines the sharpness of transition between the maximum and minimum value of $p_E$.

## Simulation details

We numerically solved the ODEs (Equation 1) using MATLAB (ode45 solver). Initially, all the variables were kept at zero for the numerical solution. We took some model parameters from a previous study [68], while other parameters were obtained by fitting to the data of hypocotyl lengths from wildtype seedlings grown under 4, 8, 12, and 16-hr white light cycles at 22°C (see the details of parameter selection and fitting procedure below). Then, we predicted the hypocotyl lengths and GUS activities for other genotypes by varying the parameters, $mut_K$, $P^*$, and $p_{self}$ (as in S2 Table).

## Parameter selection and fitting

We first fixed some of the parameters from a previous study [68], on which our model is based (see S1 Table). Next, there were 7 more unknown parameters, which were estimated from the `non-linear least squares' fitting to the wild-type (WT) hypocotyl data at $22^oC$ temperature. Also, note that all multiplicative factors for the WT are unity by definition ($mut_k = 1$).

The non-linear least squares method was implemented using the open-source Python package *scipy.optimize.curve_fit* (Version SciPy v1.12.0). To implement this method, the number of data points should be more than or at least equal to the number of unknown parameters. However, we have only four experimental data points of mean hypocotyl lengths (each averaged over ~20 replicates) and seven unknown parameters. Due to the limited number of experimental data points, we have considered all ~20 replicates to estimate the parameter uncertainty (given by standard deviation in S2 Table), and the corresponding goodness of fit (given by $R^2$ values). The $R^2$ values for fitting the model with experimental data points are shown in Fig 2B.

After determining all parameters for the WT, we did not alter some of the parameters for other genotypes, namely, inhibition rate of ELF3 by COP1 ($d_{EC}$), basal GUS production rate ($k_0$), GUS production rate influenced by PIF4($p_F$), GUS decay rate($d_F$), and intensity of PIF4's inhibition of GUS($p_{FP}$). For all mutants and overexpression lines, we varied only three parameters, PIF4 threshold concentration for autoinhibition ($P^*$), negative feedback strength ($p_{self}$), and the multiplicative factors ($mut_k$), to again estimate their values by fitting to the corresponding hypocotyl data for each genotype (see S2 Table).

Finally, note that all parameter values (with corresponding standard deviation) were obtained by fitting only for $22^oC$ temperatures. However, the values of threshold ($P^*$) and feedback strength ($p_{self}$) were assumed to change slightly at $27^oC$ temperature (without any fitting).

**Effect of variation of the PIF4 autoinhibition strength over time**

To predict the dynamics of PIF4 protein and GUS activity over a diurnal cycle (Fig 7A, 7B, and 7C), we assumed a temporal variation of the PIF4 autoinhibition strength. Since we observed the peaks in PIF4 protein levels around the middle of the day, we assumed that the autoinhibition strength is the lowest during mid-day, and it is higher in the beginning of the day and end of the night. In particular, the autoinhibition strength of PIF4 (denoted by $P_{self}$) is assumed to have the following mathematical form:

$$P_{self} = -\left(\frac{P_{self}^{max/day} - P_{self}^{min}}{D_1}\right) t_0 + P_{self}^{max/day}, \quad t_0 \leq D_1$$

$$P_{self} = P_{self}^{min}, D_1 < t_0 \leq D_2$$

$$P_{self} = \left(\frac{P_{self}^{max/night} - P_{self}^{min}}{24 - D_2}\right)(t_0 - D_2) + P_{self}^{min}, \quad t_0 \geq D_2,$$

(3)

Where, $t_0 = mod(t, 24)$, and $P_{self}$ stays minimal through the interval from $D_1 = 2\ hour$ to $D_2 = 6$ hour of a diurnal cycle. This assumed functional form of PIF4 autoinhibition strength is shown in Fig 7D.

Similarly, the inhibition strength of GUS expression by PIF4 follows the following equation

$$P_{FP} = -\left(\frac{P_{FP}^{max/day} - P_{FP}^{min}}{D_1}\right) t_0 + P_{FP}^{max/day}, \quad t_0 \leq D_1$$

$$P_{FP} = P_{FP}^{min}, D_1 < t_0 \leq D_2$$

$$P_{FP} = \left(\frac{P_{FP}^{max/night} - P_{FP}^{min}}{24 - D_2}\right)(t_0 - D_2) + P_{FP}^{min}, \quad t_0 \geq D_2$$

(4)

By incorporating these time-dependent variations of autoinhibition strengths (Equations 3 and 4) into the ODEs of our model (Equation 1), we qualitatively captured the dynamics of both PIF4 protein levels and GUS activity over day and night (Fig 7E and 7F), using some additional parameter set (see Table S3).

## Quantification and statistical analysis

In all experiments, the data are presented as mean±SD. The number of biological replicates (n) and the statistical details are indicated in the corresponding figure legends. For the hypocotyl length measurement and quantification of western blots, Image J software was used. GraphPad Prism 8.0 and Microsoft Excel were used for graph preparation and statistical analysis. The statistical significance between or among treatments and/or genotypes was determined based on one-way or two-way ANOVA, followed by Tukey's HSD test ($P < 0.05$). Significant differences between genotypes and/or temperatures are denoted by different letters. For immunoblot data, all the experiments were repeated three times, and the data from the representative experiment are shown.

## Supporting information

**S1 Fig. PIF4 negatively autoregulates its own gene expression in a photoperiod-dependent manner.**
(PDF)

**S2 Fig. Constitutive overexpression of PIF4 results in stronger autoinhibition of PIF4 promoter activity.**
(PDF)

**S3 Fig. Auto-inhibition of the PIF4 promoter persists in the adult stage.**
(PDF)

**S4 Fig. COP1/DET1 promotes PIF4 autoinhibition.**
(PDF)

**S5 Fig. The phyB photoreceptor inhibits PIF4 autoinhibition.**
(PDF)

**S6 Fig. Temperature-mediated autoinhibition of *PIF4* promoter activity depends on endogenous PIF4 protein concentration.**
(PDF)

**S7 Fig. Hypothetical model depicting the PIF4-mediated autoinhibition.**
(PDF)

**S1 Data. Underlying numerical data for Figs 1–7 and S1-S6.**
(XLSX)

**S1 Table. The description of parameters and corresponding values.** The values are taken from [68], and they are fixed for all genotypes. Note that the shaded boxes denote the parameters that slightly change from 22°C to 27°C.
(PDF)

**S2 Table. Parameter values estimated (with uncertainty) from non-linear least squares fitting.** Note that the shaded boxes represent the parameter values obtained by fitting to the wildtype data and kept unchanged for all other genotypes.
(PDF)

**S3 Table. Parameter values to produce the dynamics of PIF4 protein and GUS activity (corresponding to Fig 6G and 6H).** Shaded boxes represent the unchanged parameter values for all genotypes.
(PDF)

**S4 Table. List of oligonucleotide sequences used in the study.**
(PDF)

## Acknowledgments

We thank Dr. Vinod Kumar for *pPIF4:PIF4-FLAG* seeds and Prof. Salome Prat for the *pPIF4:LUC* seeds. We also thank all the members of the Gangappa lab for fruitful discussions. We also thank Sumana Annagiri's lab (IISER Kolkata) for the microscopy facility.

## Author contributions

**Conceptualization:** Dipjyoti Das, Sreeramaiah N. Gangappa.

**Data curation:** Sreya Das, Vikas Garhwal, Krishanu Mondal, Dipjyoti Das.

**Formal analysis:** Sreya Das, Vikas Garhwal, Krishanu Mondal.

**Funding acquisition:** Sreeramaiah N. Gangappa.

**Investigation:** Vikas Garhwal.

**Methodology:** Sreya Das, Vikas Garhwal, Krishanu Mondal, Dipjyoti Das.

**Project administration:** Dipjyoti Das, Sreeramaiah N. Gangappa.

**Resources:** Sreya Das, Vikas Garhwal, Dipjyoti Das, Sreeramaiah N. Gangappa.

**Software:** Krishanu Mondal, Dipjyoti Das.

**Supervision:** Dipjyoti Das, Sreeramaiah N. Gangappa.

**Validation:** Sreya Das, Vikas Garhwal, Krishanu Mondal, Dipjyoti Das.

**Visualization:** Sreya Das, Vikas Garhwal, Krishanu Mondal, Dipjyoti Das.

**Writing – original draft:** Sreya Das, Vikas Garhwal, Dipjyoti Das, Sreeramaiah N. Gangappa.

**Writing – review & editing:** Sreya Das, Vikas Garhwal, Dipjyoti Das, Sreeramaiah N. Gangappa.

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
