## [Decision Letter · Decision Letter 0]

PGENETICS-D-24-01351

Threshold-dependent negative autoregulation of PIF4 gene expression optimizes growth and fitness in Arabidopsis

PLOS Genetics

Dear Dr. Gangappa,

Thank you for submitting your manuscript to PLOS Genetics. After careful consideration, we feel that it has merit but does not fully meet PLOS Genetics's publication criteria as it currently stands. Therefore, we invite you to submit a revised version of the manuscript that addresses the points raised during the review process.

Please submit your revised manuscript within 60 days Apr 20 2025 11:59PM. If you will need more time than this to complete your revisions, please reply to this message or contact the journal office at plosgenetics@plos.org. Please include the following items when submitting your revised manuscript:

We look forward to receiving your revised manuscript.

Kind regards,

Enamul Huq, Ph.D.

Guest Editor

PLOS Genetics

Monica Colaiácovo

Section Editor

PLOS Genetics

Aimée Dudley

Editor-in-Chief

PLOS Genetics

Anne Goriely

Editor-in-Chief

PLOS Genetics

**Journal Requirements:**

- TM on pages: 26, and 27.

4) We have noticed that you cited Figure S6I and S6J on page 18 in your manuscript. However, there is no corresponding file uploaded to the submission. Please include it in the Supporting Information file or remove all reference to it within the text.

Potential Copyright Issues:

i) Please confirm (a) that you are the photographer of 1C, 1F, 3A, 3F, 3K, 3L, 4A, 4F, 4J, , 5A, 5H, S1C, S1E, S1Q, S2C, S2E, S3A, S3C, S3E, S3G, S4D, S4F, S4H, S4L, S4N, S4P, S5B, S5H, and S5I, or (b) provide written permission from the photographer to publish the photo(s) under our CC BY 4.0 license.

7) Please ensure that the funders and grant numbers match between the Financial Disclosure field and the Funding Information tab in your submission form. Note that the funders must be provided in the same order in both places as well. Currently, "IISER Kolkata Intramural grant (Ministry of Education, Government of India)" is missing from the Funding Information tab.

**Reviewers' comments:**

Reviewer's Responses to Questions

Reviewer #1: This study by Das et al., report that PIF4 undergoes a threshold-dependent autoinhibition of itself at the transcriptional level. This regulatory event contributes to the growth and biomass accumulation in Arabidopsis. Although previous studies have shown that PIF4 binds to its own promoter regions to control its transcription, the detailed molecular mechanism and biological significance are not clear. Overall, this is a nice story. The conclusions claimed in this study are well-supported by the genetic and biochemical data. I only have a few suggestions.

1. Please indicate (where it is missing) how many biological replicates were used in each of the genetic and biochemical studies.

2. The working model showing in Figure 7G lacking the light (red light) and warm temperatures-regulated PIF4 accumulation and autoinhibition of its own expression.

3. Please indicate the number of plants were used to measure the rosette biomass (Figure 7C-D).

Reviewer #2: PIF4 is a central transcription factor integrating various environmental signals, including light and temperature, to regulate plant growth and development. Previous studies have shown that PIF4 can bind to its own promoter, indicating a feedback mechanism regulating its own transcription. In this manuscript, Das et al. propose an interesting model in which PIF4’s autoregulation is threshold-dependent and essential for maximising growth and fitness. I find this model quite compelling, and I believe that most of the data support it well. Below are my comments:

1. In the abstract and introduction, the authors state that the regulation of PIF4 transcription is poorly understood. However, recent studies by several groups have demonstrated that a number of transcription factors, including HSFA1, TCP5, and BBX18, regulate PIF4 transcription. Thus, the authors’ statement is not entirely accurate and should be revised. These studies also need to be incorporated into the introduction and discussed accordingly.

2. The Western blot data appear rather messy. I can see multiple bands in the wild-type, whereas most of these bands are absent in the pif4 mutant. Does this imply that all of these bands represent PIF4? Is this situation typical?

Apart from these points, I have no further questions.

Reviewer #3: This study shows how PIF4 autoinhibits its own promoter activity in a concentration and photoperiod dependent manner. The authors have used both experimental analysis and mathematical modelling to try substantiating their hypothesis. They have used PIF4 mutant and overexpressor lines as well as different pPIF4-GUS lines in different genetic backgrounds for the study. They observed that PIF4 promoter activity is increased in pif4, whereas it is reduced in PIF4 overexpression lines. Authors also bring in the possible role of phyB and COP1 in mediating PIF4 autoinhibition. While the study is interesting, I have some reservations about the claims in the study which lack proper experimental evidence and some other comments that are given below.

Major Comments:

1. The authors have used the term ‘threshold concentration’ several times in the manuscript. What is the threshold concentration?

2. Page no: 13 “However, an exact quantitative comparison of the GUS activity with experimental data cannot be performed since our model's GUS activity is measured in arbitrary units.” Please elaborate on how the GUS activity is measured using the mathematical model in “arbitrary units”.

3. In Figure 1 b and d, according to authors, PIF4 expression is more in overexpressor than wt but promoter activity is less. The overexpresssor (pPIF4:PIF4-FLAG) authors used were under native promoter. Kindly explain this observation.

4. Figure 1 n, authors showed the GUS activity in red light, one experiment with gus activity comparison between red light and white light in wt is needed here just to show that what is the effect of red light on PIF4 promoter activity in comparison to white light.

5. Reason for PIF4 promoter activity reduction in pif4-101 mutant under red light is not explained properly.

6. What is the rationale behind choosing ZT23 as the timepoint for harvesting seedlings for the experiments?

7. Under SD condition, at ZT23, when PIF4 mRNA, protein and its activity all are at the peak to promote hypocotyl elongation, then it needs a better experimental clarification why PIF4OE1 (which is driven by PIF4 promoter) showing higher transcript level but the introgressed line with pPIF4:GUS showing reduced promoter activity compared to wild-type. Kindly explain.

8. Author can provide the western for CCA1/LHY1 or SHB1 at SD ZT23 especially in cop1 mutant background, which are the known positive regulators of PIF4 expression. Also, in cop1 mutants we would expect high abundance of HY5 in the nucleus. HY5 inhibits the targets of PIF4. Because of these factors won’t we expect promoter activity in cop1 mutant even if PIF4 is not stable. No visible GUS stain in cop1 background compared to WT and pif4-101 needs further experimental evidence.

9. Fig 3A and 3B: Authors have observed that PIF4 promoter activity as well as PIF4 transcript levels are significantly reduced in 35S:COP1 than in WT, due to PIF4 concentration dependent autoinhibition. Since COP1 positively regulate PIF4 promoter activity, there should be a threshold PIF4 concentration below which COP1 positive regulation dominates more than the PIF4 mediated autoinhibition resulting in an increased pPIF4-GUS activity. In this regard, was the GUS activity measured in seedlings younger than 6 days old?

10. The justification behind including rosette biomass and seed yield in the manuscript without enough experimental support is not clear.

Minor Comments:

1. In introduction last paragraph, many sentences are redundant, kindly reframe the paragraph.

2. In Figure 5I, kindly check the statistics of 35S:PHYB in 22’C and 27’C, they do not seem to be in the same statistical group.

3. In Figure 6 A, WT 22 and WT 28, protein size of PIF4 band is looking different. Kindly check carefully.

4. In all the blots pif4 is not showing the non-specific bands seen in Col-0, why?

5. In Fig S1O ZT23 is chosen for SD and in Figure S1M, ZT4 was chosen for SD, kindly explain.

6. What is the COP1 expression level in cop1 mutant and overexpressor crossed with pPIF4:GUS?

7. Figure 6 legend: “All the immunoblotting experiment was performed at teh same time and maintained the identical conditions thrpughout the procedure.” Kindly check and correct the spelling.

8. Figure 7A and B: How many days old seedlings were harvested in order to check the relative expression levels of the temperature induced and growth responsive genes YUC8 and IAA29 ?

9. Why was PIF4 particularly chosen for the study? Were any other PIFs as well checked for their role in autoregulation similarly to PIF4?

**Have all data underlying the figures and results presented in the manuscript been provided?**

Reviewer #1: Yes

Reviewer #2: Yes

Reviewer #3: None

PLOS authors have the option to publish the peer review history of their article (what does this mean? ). If published, this will include your full peer review and any attached files.

**Do you want your identity to be public for this peer review?** For information about this choice, including consent withdrawal, please see our Privacy Policy .

Reviewer #1: **Yes: ** Dongqing Xu

Reviewer #2: **Yes: ** Yongjian Qiu

Reviewer #3: No

**Figure resubmission:**
---

## [Decision Letter · Decision Letter 1]

Dear Dr Gangappa,

p.p1 {margin: 0.0px 0.0px 0.0px 0.0px; font: 9.0px Helvetica; color: #262

We are pleased to inform you that your manuscript entitled "Threshold-dependent negative autoregulation of PIF4 gene expression optimizes growth and fitness in Arabidopsis" has been editorially accepted for publication in PLOS Genetics. Congratulations!

Yours sincerely,

Enamul Huq, Ph.D.

Guest Editor

PLOS Genetics

Monica Colaiácovo

Section Editor

PLOS Genetics

Aimée Dudley

Editor-in-Chief

PLOS Genetics

Anne Goriely

Editor-in-Chief

PLOS Genetics

Comments from the reviewers (if applicable):

Reviewer's Responses to Questions

**Comments to the Authors:**

Reviewer #1: The authors have fully addressed my concerns.

Reviewer #3: The authors have modified the manuscript as suggested. They gave a point-by-point response to all our queries and also to other reviewers', so I don't have any further questions.

**Have all data underlying the figures and results presented in the manuscript been provided?**

Reviewer #1: Yes

Reviewer #3: None

PLOS authors have the option to publish the peer review history of their article (what does this mean? ). If published, this will include your full peer review and any attached files.

**Do you want your identity to be public for this peer review?** For information about this choice, including consent withdrawal, please see our Privacy Policy .

Reviewer #1: No

Reviewer #3: No

**Data Deposition**

http://datadryad.org/submit?journalID=pgenetics&manu=PGENETICS-D-24-01351R1

**Press Queries**

---

## [Editor Report · Acceptance letter]

PGENETICS-D-24-01351R1

Threshold-dependent negative autoregulation of PIF4 gene expression optimizes growth and fitness in Arabidopsis

Dear Dr Gangappa,

We are pleased to inform you that your manuscript entitled "Threshold-dependent negative autoregulation of PIF4 gene expression optimizes growth and fitness in Arabidopsis" has been formally accepted for publication in PLOS Genetics! Your manuscript is now with our production department and you will be notified of the publication date in due course.

With kind regards,

Judit Kozma

PLOS Genetics

On behalf of:
